# A new seasonal frozen soil water-thermal coupled migration model and its numerical simulation

**Chaoyi Zhang**[1], **Feng Chen**[2], **Lei Sun**[1]*, **Zhangchao Ma**[1], **Yan Yao**[3]

**1** Automation and Electronic Engineering College, University of Science and Technology Beijing, Beijing, China, **2** Forestry Institute, Beijing Forestry University, Beijing, China, **3** Automation College, Beijing University of Posts and Telecommunications, Beijing, China

* sun_lei@ustb.edu.cn

**Data Availability Statement:** All relevant data are within the manuscript and its Supporting Information files.

**Funding:** Beijing Natural Science Foundation (6194041), the National Key R&D Program of

## Abstract

In this paper, a mathematical model based on spherical differential unit cell is proposed as a model for studying seasonal freeze-thaw soil space infinitesimal differential unit cell. From this model, the basic equations of permafrost moisture and heat flow motion are directly derived, then the linked equations form the permafrost water-heat coupled transport model. On this basis, the one-dimensional seasonal permafrost water-heat transport equation is derived. The model reduces the original spatial three-variable coordinate system (parallel hexahedron) into a coupled equation with a single spherical radius (R) as the independent variable, so the iterations of the numerical simulation algorithm is greatly reduced and the complexity is decreased. Finally, the model is used to simulate the seasonal freeze-thaw soil in the ShiHeZi region of Xinjiang, China. The principle of the simulation is to collect the soil temperature and humidity values of the region in layers and fixed-points using a home-made freeze-thaw soil sensor, after that we solve it by numerical calculation using MATLAB. The analysis results show that the maximum relative error of the model we proposed is 4.36, the minimum error is 0.98, and the average error is 2.515. The numerical simulation results are basically consistent with the measured data, then the proposed model is consistent with the matching states of permafrost moisture content and soil temperature in the region at different times. In addition, the experiments also demonstrate the reliability and accuracy of the model.

## Introduction

With regard to the recent progress on permafrost research, the results are mainly as follows: ① During a complete annual freeze-thaw cycle, all layers of near-surface soils has generally experienced four stages: summer thawing period, spring and autumn thawing-freezing period, as well as winter freezing period. Depending on local factors, there are differences freezing and thawing characteristic in different regions, such as the start and end time, the frozen rate, the soil type, etc. [1–3]. ② The difference of daily freeze-thaw cycle between permafrost region

China (2020YFB1708800), Fundamental Research Funds for the Central Universities(FRF-MP-20-37).

**Competing interests:** The authors have declared that no competing interests exist.

and seasonal frozen region is large, which is mainly reflected in the duration of daily freeze-thaw cycle [4, 5]. ③ Different land surface models can well grasp the spatial and temporal variation of physical quantities in freeze-thaw processes, but all need to be parameterized and improved according to the characteristics of land surface processes [6, 7]. ④ Based on the thermodynamic equilibrium equation, circumventing the unstable iterative calculation and determining the freeze-thaw critical temperature can improve the unreasonable freeze-thaw parameterization scheme [8].

Literatures [9, 10] considered the unfrozen water content in the cold region to improve the accuracy of coupling simulation of heat transfer and water in frozen soil. The comprehensive algorithm and parameterization were used to calculate the thaw water content, also the selected parameters of unfrozen water content were evaluated by using soil temperature, specific surface area of soil particles, soil water curve and different types of water. The final results showed that the parameterization of unfrozen water content was affected by many factors, the heating and cooling process was particularly important when calculating the unfrozen water content. The existing problems was [9–11]: there were many physicochemical parameters in these studies, which were not easy to obtain. They were difficult to be used to calculate unfrozen water content. A practical high-precision physicochemical parameter needed to be developed to couple with freezing model and land surface process.

Literatures [12–14] considered it crucial to identify modeling schemes for macropore-matrix interactions and permeate water refreezing, discussed the necessity to study the effects of macropore flow and soil freeze-thaw interactions, the need to integrate these concepts into a framework of coupled hydrothermal transfer, then they proposed a conceptual model for freezing unsaturated flow in macropore soils that assumed two interacting domains (macropore and matrix) with different water and heat transfer mechanisms. Existing problems were [12, 15, 16]: the detailed understanding of macropore flow mechanism in permafrost, and how it changes with different soil thermal conditions were still uncertain in these proposed models, so it was necessary to further develop the existing macropore flow description and various scale modeling methods. New modeling approached that test these concepts and quantify these dynamics can address the rate of water flow in frozen micropore soils and investigate the conditions that allow water to bypass the freezing zone, or cause water to freeze in the macropores in the opposite way.

In terms of technical means, literature [17] used the interferometric synthetic aperture radar (InSAR) technology to monitor the surface of permafrost area on a large scale all day and all-weather, summarized the application of D-InSAR and time series InSAR technology in permafrost area, analyzed the factors affecting the surface deformation in permafrost area. Existing problems were [17–19]: under the global warming, the accumulated long time series SAR data were fully utilized to achieve the multi-faceted research effect of different scales to study the law of multi-year permafrost change. The continuous collection and expansion of field measurement data will improve the existing freeze-thaw models, while different physical parameter models will be developed to supplement the gaps in this field, this was the main trend of future development of permafrost measurement technology.

In literature [20], thermal properties of soils and various other physical properties were determined by using the heat pulse (HP) method, which was based on a linear heat source solution of radial heat flow equation, a high-pressure probe structure was proposed, the properties measured in unfrozen and frozen soils were discussed. The problems with these frozen soil observation techniques [17–23] were: the current probe design limits the representative volume of soil samples, its extreme sensitivity to needle distance leads to the lack of accuracy and durability. Moreover, the short length of thermal TDR needle affects the accuracy and precision of moisture or water flux estimation. New theories, methods, heating strategies and

probe designs for estimating soil thermal properties, unfrozen water content and ice content were still in infancy, but they might greatly improve our understanding of measurement techniques. In addition, available thermal conductivity models developed from steady-state measurements need to be evaluated and calibrated.

In this paper, the water and heat transfer process of permafrost microcell is analyzed, a spherical spatial coordinate system is proposed as the research model by using the spatial rectangular coordinate system of parallelogram differential cell. In the rectangular coordinate system, the cubic model of previous study is replaced by the spherical model. This is because the spherical model is more consistent with the general law of moisture motion and water vapor in terms of distribution in soil space, while the edges and corners of the cube model have errors in the random flow of water. Therefore, the spherical model is used to derive the hydrothermal coupling equation in this paper. As for the choice of coordinate system is rectangular coordinate system, because the radius and angle of spherical coordinate system, finally can be converted into the spatial correspondence of rectangular coordinate system, so this paper focuses on the minimum spatial physical unit of soil moisture movement, which is a model of spherical structure.

Under frozen conditions, based on soil water movement and heat flow equations, this study rebuilt the moisture migration equation and heat transfer equation, frozen soil hydrothermal coupled migration partial differential equations were derived using the contact equation. This model considerably simplified computational complexities, as the original three variables in three-dimensional space were reduced to only one spherical radius ($R$) independent variable in the coupled equations, thereby substantially reducing coupling iteration computations. Next, the numerical simulation method was used, and a fully implicit finite difference scheme was adopted, the time step was adjusted according to the soil temperature and moisture, the number of iterations was reduced, and the result was quickly calculated. Finally, taking seasonal frozen soil in the ShiHeZi area of Xinjiang Province as an example, results using collected data were compared with those from simulations to demonstrate the robustness and precision of the proposed model.

## Mathematical model

A homogeneous vertical one-dimensional seasonal frozen soil water and heat coupling problem was considered. Frozen soil was assumed to be under the conditions of the basic hypothesis specified in [24, 25], according to the principle of unsaturated soil water dynamics and related theory, based on which a seasonal frozen soil water and heat coupled migration partial differential mathematical model was built [26].

### A. Seasonal frozen soil water movement

Based on space rectangular coordinate system, a point $(R,\theta)$ was taken in the space of frozen soil moisture flow, $(R,\theta)$ was the centre of an infinitesimal differential unit (rigid spheres), and the radius was $\Delta R$ (Fig 1). Based on the cube model introduced in previous literature, this paper proposes the physical process of water movement under sphere model. The biggest advantage is that the particle can make any trajectory in the 360 degree free space. As a water molecule, it can enter and exit the spherical structure in the 360 degree free space. In the past, the cube micro unit can also analyze and study the movement of water molecules, but there are "edges" and "angles" in the cube, so it is not easy to calculate the movement of water molecules when passing through the edges and angles, while the sphere structure has no "edges" and "angles", which fully ensures that water can move in any direction, which is more in line with the actual situation, Therefore, this paper proposes such a new spherical micro unit for

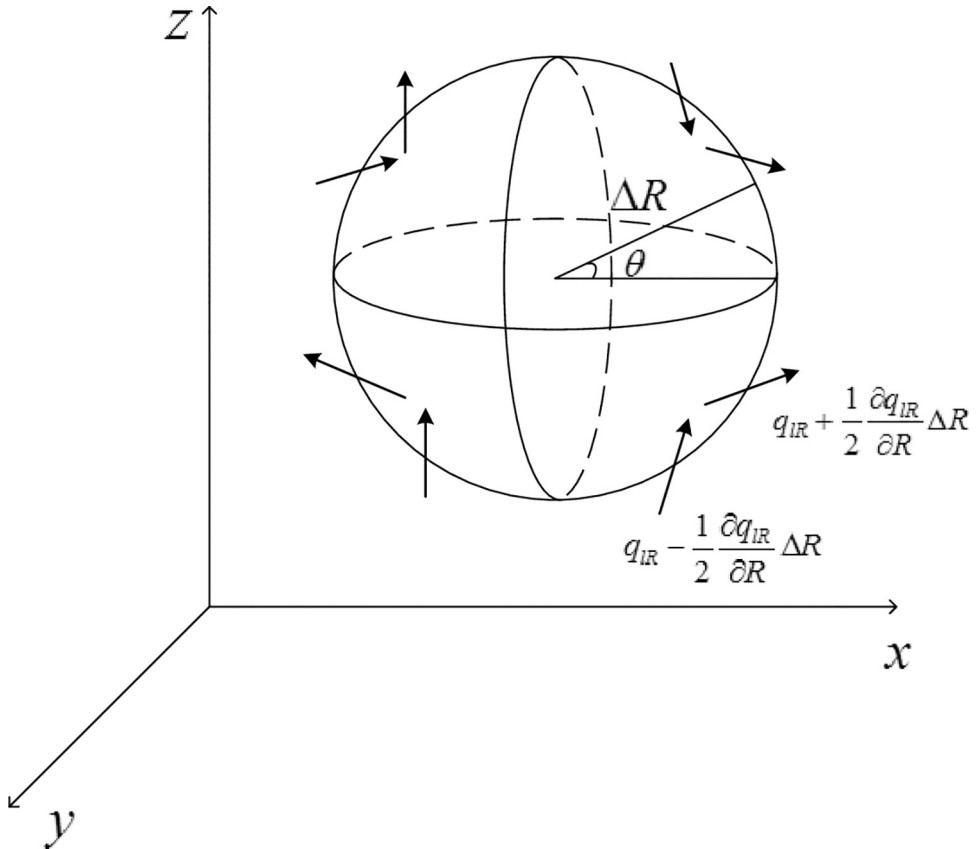

**Fig 1. Differential element in rectangular coordinates.**

induction, which is the basis of analyzing the law of water movement. Along the sphere's surface, outward and inward soil liquid moisture flux was $q_{lR}$ in the radius $R$'s direction, the density of water was $\rho_l$, the density of ice was $\rho_i$, and direction $R$'s quantity of water vapour flux was $q_{vR}$. In the time period ($\Delta t$), this model's water movement basic differential equation was built based on the law of conservation of mass.

First, liquid moisture flux flowing from the sphere's exterior to interior was given by $q_{lR} - \frac{1}{2}\frac{\partial q_{lR}}{\partial R}\Delta R$, while that from the sphere's interior to exterior was given by $q_{lR} + \frac{1}{2}\frac{\partial q_{lR}}{\partial R}\Delta R$ (unit: cm/s), such that within a small time slice $\Delta t$, inflow and outflow of the spherical differential unit cell's liquid moisture flux rate was as follows:

$$q_{lR} + \frac{1}{2}\frac{\partial q_{lR}}{\partial R}\Delta R - (q_{lR} - \frac{1}{2}\frac{\partial q_{lR}}{\partial R}\Delta R) = \frac{\partial q_{lR}}{\partial R}\Delta R \tag{1}$$

Based on Eq (1), the unit interval time of inflows and outflows of the sphere's liquid water volume change was:

$$\frac{\partial q_{lR}}{\partial R}\frac{4}{3}\pi\Delta R^3\Delta t \tag{2}$$

Denoting the density of soil moisture as $\rho_l$ and the mass density formula as $m = \rho V$, the unit interval time of water quantity change of the differential element was:

$$\frac{\partial(\rho_l q_{lR})}{\partial R}\frac{4}{3}\pi\Delta R^3\Delta t \tag{3}$$

Eq (3) was the change in a one-dimensional direction's liquid water quality, such that the total liquid water quantity change under three-dimensional coordinates was:

$$-4\frac{\partial(\rho_l q_{lR})}{\partial R}\pi\Delta R^3\Delta t \tag{4}$$

Similarly, water vapour flux flowing from the sphere's exterior to interior was $q_{vR} - \frac{1}{2}\frac{\partial q_{vR}}{\partial R}\Delta R$, while that from the sphere's interior to exterior was $q_{vR} + \frac{1}{2}\frac{\partial q_{vR}}{\partial R}\Delta R$ (unit: cm/s), such that within a small time slice $\Delta t$, inflow and outflow of the spherical differential unit cell's water vapour flux rate was as follows:

$$q_{vR} + \frac{1}{2}\frac{\partial q_{vR}}{\partial R}\Delta R - (q_{vR} - \frac{1}{2}\frac{\partial q_{vR}}{\partial R}\Delta R) = \frac{\partial q_{vR}}{\partial R}\Delta R \tag{5}$$

By liquid water volume and density formula calculation, the unit interval time of water vapour quantity change of the differential element was:

$$-4\frac{\partial(q_{vR})}{\partial R}\pi\Delta R^3\Delta t \tag{6}$$

Based on Eqs (4) and (6), the total water quantity change in unit time ($\Delta t$) was:

$$-4\frac{\partial(\rho_l q_{lR})}{\partial R}\pi\Delta R^3\Delta t - 4\frac{\partial(q_{vR})}{\partial R}\pi\Delta R^3\Delta t \tag{7}$$

Written in the form of the *nabla* operator, this would be:

$$-4[\nabla(\rho_l q_l) + \nabla(q_v)]\pi\Delta R^3\Delta t \tag{8}$$

Next, the water and ice content rate of frozen soil were taken into consideration. Liquid water quantity in the frozen soil differential unit was $\rho_l v_l \frac{4}{3}\pi\Delta R^3$, and $v_l$ was the liquid water volume (not the frozen water volume in frozen soil).Similarly, the ice quantity in frozen soil differential unit was $\rho_i v_i \frac{4}{3}\pi\Delta R^3$, where $v_i$ was the ice volume (ice volume in frozen soil). Therefore, within the rigid sphere differential unit, the total water quantity change in ($\Delta t$) (including water and ice in frozen soil) was:

$$4\frac{\partial\rho_l v_l + \rho_i v_i}{\partial t}\pi\Delta R^3\Delta t \tag{9}$$

Finally, according to the law of conservation of mass, the water mass balance equation for the frozen soil spherical differential unit was:

$$4\frac{\partial\rho_l v_l + \rho_i v_i}{\partial t}\pi\Delta R^3\Delta t = -4\frac{\partial(\rho_l q_{lR})}{\partial R}\pi\Delta R^3\Delta t - 4\frac{\partial(q_{vR})}{\partial R}\pi\Delta R^3\Delta t \tag{10}$$

Eq (10) was contracted, reduced, and written in the following operator form:

$$\frac{\partial(\rho_l v_l + \rho_i v_i)}{\partial t} = -[\nabla(\rho_l q_l) + \nabla(q_v)] \tag{11}$$

Based on the above described assumption, the density of water and ice was invariant ($\rho_l$ and $\rho_i$ were constant), and Eq (11) can be written as:

$$\rho_l \frac{\partial v_l}{\partial t} + \rho_i \frac{\partial v_i}{\partial t} = -4 \frac{\partial(\rho_l q_{lR})}{\partial R} - 4 \frac{\partial(q_{vR})}{\partial R} \rightarrow \frac{\partial v_l}{\partial t} + \frac{\rho_i}{\rho_l} \frac{\partial v_i}{\partial t} = -4 \frac{\partial(q_{lR})}{\partial R} - \frac{4}{\rho_l} \frac{\partial(q_{vR})}{\partial R} \quad (12)$$

Combined with the unsaturated soil water movement under Darcy's law, the equation becomes

$$q_l = -K(v_l)\nabla\psi = -K(v_l)\nabla(\psi_m + \psi_g) \quad (13)$$

Where $q_l$ is the liquid water flux in frozen soil, $K(v_l)$ is the unsaturated hydraulic conductivity when the liquid water content is variable in frozen soil, and $\psi$ is the unsaturated soil water potential. The equation consists of a matric potential ($\psi_m$) and gravity potential ($\psi_g$), i.e., $\psi = \psi_m + \psi_g$. Eq (13) was placed into Eq (12), and we get:

$$\frac{\partial v_l}{\partial t} + \frac{\rho_i}{\rho_l}\frac{\partial v_i}{\partial t} = 4\nabla[K(v_l)\nabla\psi] - \frac{4}{\rho_l}\frac{\partial(q_{vR})}{\partial R} \rightarrow \frac{\partial v_l}{\partial t} + \frac{\rho_i}{\rho_l}\frac{\partial v_i}{\partial t} = 4\frac{\partial}{\partial x}[K(v_l)_R \frac{\partial\psi}{\partial R}] - \frac{4}{\rho_l}\frac{\partial(q_{vR})}{\partial R} \quad (14)$$

The soil water potential of unsaturated frozen soil is composed of matric and gravity potentials. To simplify, geopotential $\psi_g$ can be taken in the gravity axis direction ($z$ axis), such that $\psi = \psi_m + z$, $z$ is positive in the upward direction and negative in the downward direction, and Darcy's law in Eq (13) can be written as:

$$q_l = -K(v_l)\nabla\psi = -K(v_l)\nabla(\psi_m + \psi_g) = -3K(v_l)\nabla(\psi_m) - K(v_l)_z \quad (15)$$

Soil was homogeneous in all directions; therefore, $K(v_l)_R = K(v_l)_z = K(v_l)$. Then, the partial differential equation of frozen soil water movement was:

$$\frac{\partial v_l}{\partial t} + \frac{\rho_i}{\rho_l}\frac{\partial v_i}{\partial t} = 4\frac{\partial}{\partial x}[K(v_l)\frac{\partial\psi_m}{\partial R}] + \frac{\partial K(v_l)}{\partial z} - \frac{4}{\rho_l}\frac{\partial(q_{vR})}{\partial R} \quad (16)$$

Then, based on Fick's law of gas, the water vapour quantity flux of frozen soil can be represented as:

$$q_{vR} = -D_{vR}\frac{\partial\rho_{vR}}{\partial R} \quad (17)$$

where $D_v$ is the water vapour diffusion rate, and $\rho_v$ is the water vapour density of interspaces in frozen soil. To sum up, the basic Eq (16) for frozen soil moisture movement can be written as:

$$\frac{\partial v_l}{\partial t} + \frac{\rho_i}{\rho_l}\frac{\partial v_i}{\partial t} = 4\frac{\partial}{\partial x}[K(v_l)\frac{\partial\psi_m}{\partial R}] + \frac{\partial K(v_l)}{\partial z} - \frac{4}{\rho_l}[\frac{\partial}{\partial x}(D_{vR}\frac{\partial\rho_{vR}}{\partial R})] \quad (18)$$

Eq (18) is the partial differential equation for frozen soil water flow.

## B. Seasonal frozen soil heat flux

There is continuous heat transfer due to solar radiation, snow cover, the soil interior, and exchanges between the soil and atmosphere. The specific situation of heat transfer can be described by temperature. In frozen soil, owing to the freezing, melting and evaporation of water, heat is transmitted, and soil temperature changes accordingly [27, 28]. Thus, soil temperature taken as a dependent variable is the main contributor to frozen soil heat flux.

The spherical differential unit was taken as shown in Fig 1; in the time interval $\Delta t$, the inflow and outflow heat rate of the microsphere included three aspects: the axis of heat

conduction, the inflow and outflow liquid quantity of heat, and the latent heat of soil water evaporation. This study, in accordance with the law of conservation of energy, and on the basis of frozen soil heat flux equation derivation [29, 30], rebuilt the seasonal frozen soil heat flux balance equation under the model presented in Fig 1 model as:

$$C_s \frac{\partial T}{\partial t} + L_v \frac{\partial \rho_v}{\partial t} - L_f \rho_i \frac{\partial v_i}{\partial t} = -3[(\frac{\partial q_{TR}}{\partial R}) + c_l \rho_l (\frac{\partial (q_{lR} T)}{\partial R}) + L_v(\frac{\partial q_{vR}}{\partial R})] \tag{19}$$

Where $q_{TR}$ was the thermal circulation along the sphere's outer and inner surfaces in radius $R$'s direction; $C_s$ was the soil heat capacity, with a unit of $J/m^3 \cdot {}^o C$; $L_v$ was water's latent heat of vaporization, with a unit of $J/kg$; $\rho_v$ was the water vapour density; $L_f$ was ice's melting latent heat, with a unit of $J/kg$; and $c_l$ was the specific heat of liquid water. According to the law of Fourier heat conduction, we get:

$$q_{TR} = -\lambda_R \frac{\partial T}{\partial R} \tag{20}$$

Where $\lambda_R$ is the thermal conductivity in each direction of soil. Using the soil water potential formula in section A ($\psi = \psi_m + z$), combined with Darcy's law of unsaturated soil water movement, the basic equation of frozen soil heat flux can be given as:

$$C_s \frac{\partial T}{\partial t} + L_v \frac{\partial \rho_v}{\partial t} - L_f \rho_i \frac{\partial v_i}{\partial t}$$
$$= 3[\frac{\partial}{\partial R}(\lambda_R \frac{\partial T}{\partial R})] + c_l \rho_l (\frac{\partial (TK_z)}{\partial R}) + 3c_l \rho_l [\frac{\partial}{\partial R}(TK_R \frac{\partial \psi_m}{\partial R})] - L_v(\frac{\partial q_{vR}}{\partial R})] \tag{21}$$

Written in operator form, we obtain the following:

$$C_s \frac{\partial T}{\partial t} + L_v \frac{\partial \rho_v}{\partial t} - L_f \rho_i \frac{\partial v_i}{\partial t} = \nabla(\lambda \nabla T) + \frac{\partial (TK_z)}{\partial R} + c_l \rho_l \nabla(TK \nabla h) - L_v \nabla(\bar{q}_v) \tag{22}$$

Each direction of soil was homogeneous, the coefficient of thermal conductivity was $\lambda_R = \lambda$, and the unsaturated hydraulic conductivity was $K_z = K$; thus, the partial differential equation of frozen soil heat flow movement can be given as

$$C_s \frac{\partial T}{\partial t} + L_v \frac{\partial \rho_v}{\partial t} - L_f \rho_i \frac{\partial v_i}{\partial t}$$
$$= 3[\frac{\partial}{\partial R}(\lambda \frac{\partial T}{\partial R})] + c_l \rho_l (\frac{\partial (TK)}{\partial R}) + 3c_l \rho_l [\frac{\partial}{\partial R}(TK \frac{\partial \psi_m}{\partial R})] - L_v(\frac{\partial q_{vR}}{\partial R})] \tag{23}$$

## C. Seasonal frozen soil hydrothermal coupling equation

The frozen soil hydrothermal coupling equation was the combination of section A's movement equation and section B's heat-flow equation. The equation was used to describe the distribution of water in frozen soil. However, Eqs (18) and (23) contained three unknown variables: $v_l(R,t)$, $v_i(R,t)$, and $T(R,t)$. It is necessary to create a contact equation, i.e., the non-frozen water's relationship between moisture content and temperature in frozen soil. In the soil, the non-frozen water content kept a dynamic balance with temperature, the relationship was the characteristic curve of frozen soil, and the representative relationship between soil moisture and heat condition was given by:

$$v_l \leq v_m(T) \tag{24}$$

Where $v_m(T)$ was the largest non-frozen water content of frozen soil under the condition of negative temperature. For frozen soil, $v_i > 0$, and Eq (24) has an equal sign.

In addition, the soil temperature gradient was also one of the important factors affecting soil's water flow. After including the factor affecting the soil temperature gradient, the unsaturated soil water movement of Darcy's law can be expressed as follows:

$$q_{lR} = -D(v_l)_R \frac{\partial v_l}{\partial R} - D_{TR} \frac{\partial T}{\partial R}$$ (25)

Where $D_{TR}$ was the diffusion rate caused by the temperature potential gradient of soil's liquid water along each direction of the sphere model on radius $R$. Using Eq (25) in water movement Eq (18), the final seasonal frozen soil water heat coupling equation can be given as:

$$\frac{\partial v_l}{\partial t} + \frac{\rho_i}{\rho_l} \frac{\partial v_i}{\partial t} = 4 \frac{\partial}{\partial x} \left[ D(v_l)_R \frac{\partial v_l}{\partial R} + D_{TR} \frac{\partial T}{\partial R} \right] + \frac{\partial K}{\partial z} - \frac{4}{\rho_l} \left[ \frac{\partial}{\partial x} \left( D_{vR} \frac{\partial \rho_{vR}}{\partial R} \right) \right]$$ (26)

The seasonal frozen soil hydrothermal coupling equation (Eq (26)), frozen soil heat flux movement equation (Eq (23)) and contact equation (Eq (24)) were combined as simultaneous equations. These equations constituted the model of thermal coupling migration of seasonal freezing and thawing soil water.

## D. One-dimension seasonal frozen soil hydrothermal coupling migration

Before proposing a one-dimensional seasonal frozen soil hydrothermal coupling migration equation, some conditions are assumed: ① The soil medium was assumed to be incompressible, homogeneous and isotropic; ② Only liquid water can move in frozen soil, while ice is stationary; ③ The influence of water vapour migration acting on non-frozen water and heat flux in freezing and thawing soil was ignored; ④ Moisture migration caused by the temperature gradient was ignored; ⑤ Moisture movement caused by heat flow was ignored; ⑥ Non-frozen water content of frozen soil and soil negative temperature were in a dynamic balance state; ⑦ It can be suggested that water and heat migration mainly occur in the vertical direction; thus, the frozen soil water thermal transport problem can approximate the one-dimensional vertical problem.

**(1) One-dimensional frozen soil water, heat transport equation.** Generally, the basic water movement rule of frozen soil is similar to that of unsaturated soil water movement, and the influence of ice on water's flow is small in soil. Based on the phase transition of the Richards equation [4], constructed water transport equation in frozen soil. Phase transition refers to the process of a substance changing from one phase to another. The physical and chemical properties of the substance system are completely the same. The homogeneous part with obvious interface with other parts is called phase, which corresponds to solid, liquid and gas states. The substance has solid phases, liquid phases and gas phases. The soil phase transition mainly refers to the heat absorbed or released by a certain amount soil, such as the soil micro unit in Fig 1, when it changes from one phase to another under certain conditions (such as constant temperature). There are three kinds of heat: evaporation heat (from liquid phase to gas phase), melting heat (from solid phase to liquid phase) and sublimation heat (from solid phase to gas phase). Lei Zhidong et al. [7] considered that the Richards equation only considered the effect of water migration and phase transition, but not the effect of heat conduction. However, the other scholars [9–16] considered the heat conduction and phase change latent heat, but did not consider the water migration. On the basis of these two equations, Lei Zhidong et al. [7] put forward the coupled water and heat transfer equation of frozen soil, which considered the factors of soil water transfer, heat conduction, water phase change latent heat and so on.

In this paper, the physical parameters of frozen and unfrozen soil in severe cold area are tested. The expression of natural temperature field of soil and atmosphere is used to analyze

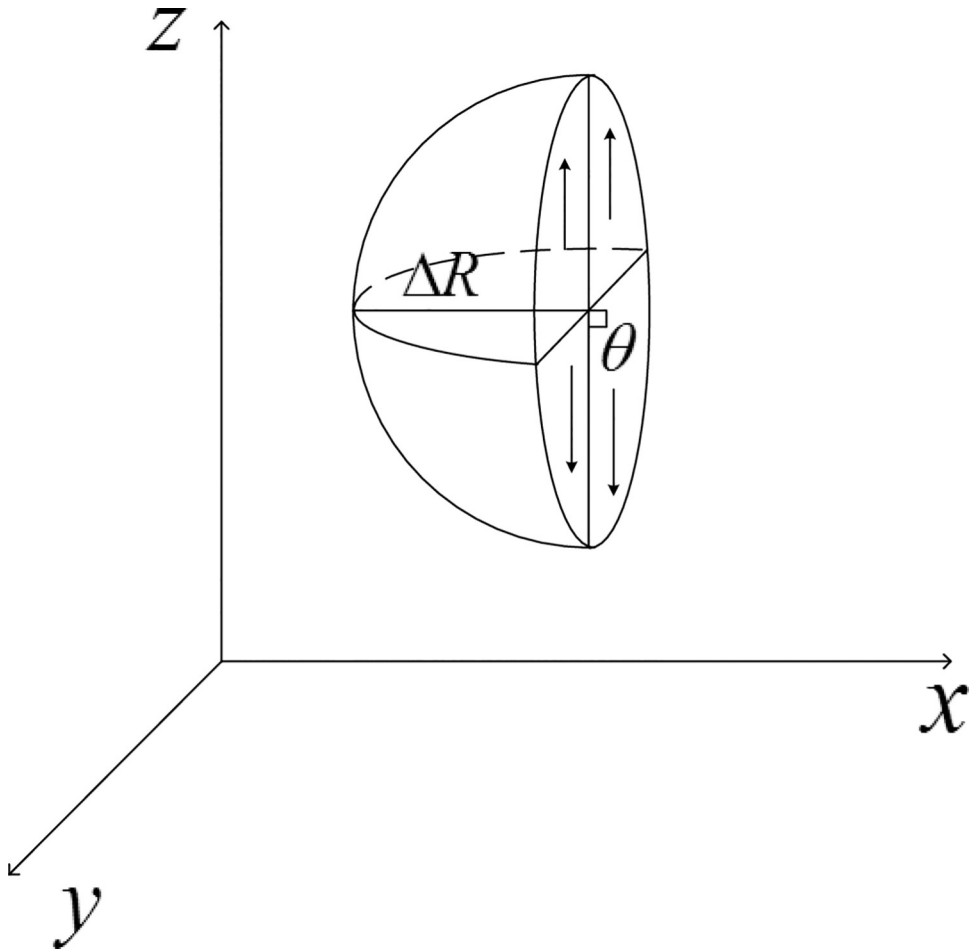

**Fig 2. One-dimension frozen soil moisture and heat coupling model.**

the influence of soil freezing and unfrozen physical parameters on soil's natural temperature field at different depths, but the phase transformation process of soil freezing and unfrozen state is not considered. Mixed-type and $\theta$ Richards one-dimensional frozen soil moisture migration can be expressed as follows (Fig 2's model):

$$\frac{\partial v_l}{\partial t} = \frac{\partial}{\partial R}\left[k(v_l)\frac{\partial \psi}{\partial R}\right] - \frac{\partial k(v_l)}{\partial R} - \frac{\rho_i}{\rho_l}\frac{\partial v_i}{\partial t} \tag{27A}$$

$$\frac{\partial v_l}{\partial t} = \frac{\partial}{\partial R}\left[\mathrm{D}(v_l)\frac{\partial v_l}{\partial R}\right] - \frac{\partial k(v_l)}{\partial R} - \frac{\rho_i}{\rho_l}\frac{\partial v_i}{\partial t} \tag{27B}$$

$$\angle\theta = -90° \tag{28}$$

Symbols were the same as those previously defined. The mixed Richards Eq (27A) was a general form of the unsaturated soil water movement equation. This equation can be used for heterogeneous soil, saturated soil and unsaturated soil moisture migration issues. It also facilitated the analysis of the hysteresis effect of soil water retention, which affects soil moisture migration. The Richards Eq (27B) was the deformation of Eq (27A), and it had the advantage

of being easily solved through numerical methods, but there were certain limitations, such as that it could only be applied to homogeneous unsaturated water movement in soil.

The one-dimensional frozen soil heat flux transport equation with phase-changing latent heat as the heat source was given by:

$$C_s \frac{\partial T}{\partial t} = \frac{\partial}{\partial R} \left( \lambda \frac{\partial T}{\partial R} \right) + L_f \rho_i \frac{\partial v_i}{\partial t} \tag{29}$$

Symbols were the same as those previously defined.

**(2) One-dimension frozen soil hydrothermal coupling equation.** Frozen soil shows strong coupling between heat and water migration, which came from moisture adsorption in soil particles and the effect of frozen soil phase transition between water and ice. For homogeneous soil, the unsaturated soil water flow Eq (27) and heat conduction Eq (29) eliminated the source-sink term that expressed the water phase-changing effect. Finally, a one-dimensional frozen soil water thermal coupling equation can be given as [31]:

$$C_s \frac{\partial T}{\partial t} = \frac{\partial}{\partial R} \left( \lambda_e \frac{\partial T}{\partial R} \right) - U_e \frac{\partial T}{\partial R} \tag{30}$$

Where, $C_e = C_s + C_1$, $\lambda_e = \lambda + D(v_l)C_1$, $U_e = C_1 \frac{dK(v_l)}{dv_l}$, and $C_1 = L_f \rho_l \frac{\partial v_m}{\partial T}$.

$C_1$, $C_e$, $\lambda_e$, and $U_e$ respectively denote the frozen soil phase-change heat capacity, equivalent volume heat capacity, equivalent coefficient of thermal conductivity, and equivalent convective velocity. Eq (30) included soil moisture migration, heat transfer and moisture phase changing factors, which influenced the soil water thermal coupling migration process. In Eq (30), the equivalent volume heat capacity of frozen soil $C_e$ contained the soil volumetric heat capacity $C_s$; it also contained the phase-changing latent heat of soil temperature, which reflected the negative feedback between soil temperature rise (drop) and moisture phase change. The equivalent thermal conductivity $\lambda_e$ not only considered the soil surface layer coefficient of thermal conductivity but also considered the latent heat migration arising from migration of water from the 'never freezing' region to frozen areas and freezing (segregated freeze) migration.

Thus, the water flow Eq (27), heat and water coupled Eq (30), contact Eq (24), initial conditions, and boundary conditions comprise the one-dimensional model of seasonal frozen soil water thermal coupling migration. The physical and mathematical models of soil heat transfer with phase change are established by using coupled soil water heat transfer equation, considering the changes of soil and atmospheric temperature field. The numerical simulation results of whether phase change is considered, and these results are compared and analyzed in soil freezing process. The results show that the curve of soil temperature with time is no longer smooth when there is phase change, and there is a gentle duration of soil temperature at the beginning of freezing and thawing, with which compared the soil without phase change; After freezing, the soil temperature with phase change is higher than that without phase change. With the increasing of depth, the temperature difference is increased. This conclusion will be proved in the later experiments. At the same time, the impedance effect of ice is not considered.

## Numerical calculation

The model equation of seasonal frozen soil water thermal coupling migration was nonlinear. The solution is very complex, and considering the uncertainty of its initial conditions and the boundary conditions, using the general solving method is not practical. Hence, at present, the numerical calculation method is commonly used. This study used the finite difference method, which is a numerical simulation method. According to the previous calculation, implicit

difference calculation for large time steps can yield a physical real solution, and the stable performance is better for a nonlinear equation; therefore, this study used a fully implicit difference scheme for solving the equations of frozen soil thermal coupling water migration.

First, the frozen soil space region and time area should be discretized. The soil with depth coordinates as the vertical axis can be divided into $N$ layers, node $i$'s coordinate was $z_i$ ($i = 0,1,\cdots,N$), and the space step was $\Delta z_i = z_i - z_{i-1}$. Similarly, the time axis coordinate was divided into $M$ periods, node $k$'s time was $t_k$ ($i = 0,1,\cdots,M$), and the time step was $\Delta t_k = t_k - t_{k-1}$. At node $i$, time $t_k$'s frozen soil moisture content, matric potential, and temperature were $v_i^k$, $\psi_i^k$, and $T_i^k$, respectively.

Second, according to Eq (30), the discrete format of the fully implicit difference equation was given as:

$$a_i v_{i-1}^{k+1} + b_i v_i^{k+1} + c_i v_{i+1}^{k+1} = h_i \tag{31}$$

The coefficients in Eq (31) were: $a_i = -\frac{2\Delta t_{k+1} D_{i-1/2}^{k+1}}{\Delta z_i (\Delta z_i + \Delta z_{i+1})}$, $c_i = -\frac{2\Delta t_{k+1} D_{i+1/2}^{k+1}}{\Delta z_{i+1} (\Delta z_i + \Delta z_{i+1})}$, $b_i = 1 - a_i - c_i$, $h_i = v_i^k - \frac{\Delta t_{k+1}}{\Delta z_i + \Delta z_{i+1}} (K_{i+1}^{k+1} - K_{i-1}^{k+1})$.

Then, when solving Eq (23), ice was assumed to stay invariant over time period $\Delta t_k$, the redundant liquid water was frozen into ice at the end of the period, and ignoring Eq (23)'s phase transformation effect, the fully implicit discrete difference equation was:

$$a_i T_{i-1}^{k+1} + b_i T_i^{k+1} + c_i T_{i+1}^{k+1} = f_i \tag{32}$$

The coefficients in Eq (32) were: $a_i = -\frac{2\Delta t_{k+1}}{\Delta z_i (\Delta z_i + \Delta z_{i+1})} \lambda_{e,i-1/2}^{k+1} - \frac{\Delta t_{k+1}}{\Delta z_i + \Delta z_{i+1}} U_{e,i}^{k+1}$, $b_i = 1 - a_i - c_i$, $f_i = C_{e,i}^{k+1} T_i^k$, $c_i = -\frac{2\Delta t_{k+1}}{\Delta z_{i+1} (\Delta z_i + \Delta z_{i+1})} \lambda_{e,i+1/2}^{k+1} + \frac{\Delta t_{k+1}}{\Delta z_i + \Delta z_{i+1}} U_{e,i}^{k+1}$.

Eqs (31) and (32) were tridiagonal equations, which can be solved using the "chasing method" [32]. Simultaneously, the frozen soil water heat coupling equation was nonlinear and coupling, which makes solving Eqs (31) and (32) independently impossible. Therefore, iterative calculation was used at each time period, and coupled Eqs (31) and (32) were repeatedly calculated until the resulting difference before and after two iterations was within a given error range. In this study, the iterative error limits of results were 0.01°C for temperature and 0.001 for moisture content.

## Initial, boundary conditions and parameters

The proposed frozen soil hydro-thermal coupling migration model and calculation accuracy needed to be verified, and numerical simulation results and experimentally measured data should be compared. Field experiments were conducted from Nov. 2017 to Apr. 2018 in Xinjiang Province, Shihezi city, NongKeSuo pit field. The geographical location of the test pit and its surrounding environment, climate, and soil conditions were in line with the characteristics of frozen soil in the representative aspects. Before numerical simulation, the experiment's initial conditions, boundary conditions and moisture characteristic parameters had been calibrated, and all test data were ensured to be within a reasonable range.

The test pit soil depth was 200 cm, the pit had a vertical down direction, the change in temperature and moisture in the upper layer of the soil (0–100 cm) was larger, the step length was 20 cm, changes in the subsoil's (100–200 cm) temperature and moisture were small, and the step length was slightly larger (30 cm). The time step was set to 10 minutes by using a sensor. Numerical simulations employed in Matlab were used to solve the model. The selected region's frozen soil hydro-thermal coupling migration was determined experimentally for the time

period of winter of 2017 to spring of 2018, The following fitting methods used approximating discrete data with analytic expression.

## A. Initial conditions

**(1) Soil initial moisture content.**   The starting time was set as Nov. 11, 2017, the soil moisture content was initially measured, and the functional relationship between soil moisture content and profile depth was established, as shown in Fig 3.

**(2) Soil initial ice content.**   When starting the numerical simulation, air temperature was higher, the experimental area did not experience permafrost, and the soil was not frozen; thus, the initial ice content could be set as $v_i(t = 0,z) = 0$.

**(3) Soil initial temperature.**   According to temperature data from the soil profile observed on Nov. 11, 2017, the soil initial temperature and depth profile curve were simulated (Nov. 18, 2017), as shown in Fig 4.

## B. Boundary conditions

**(1) Soil moisture boundary condition.**   For the upper boundary of soil moisture content, the second kind of boundary condition was chosen, i.e., the boundary condition of the rate of evaporation $E(t)$ was known, and this boundary condition can be expressed as:

$$-D(v_0)\frac{\partial v}{\partial R}\bigg|_{s=0} + K(v_0) = -E(t) \tag{33}$$

Where $v_0$ was the soil moisture content of the soil surface ($z = 0$). In the period with freezing soil, the soil surface evaporation was weak, while in the period with thawing soil, soil evaporation increased, but the duration was short. Therefore, soil evaporation was ignored in numerical simulations, and the upper boundary of frozen soil moisture movement was flux plane.

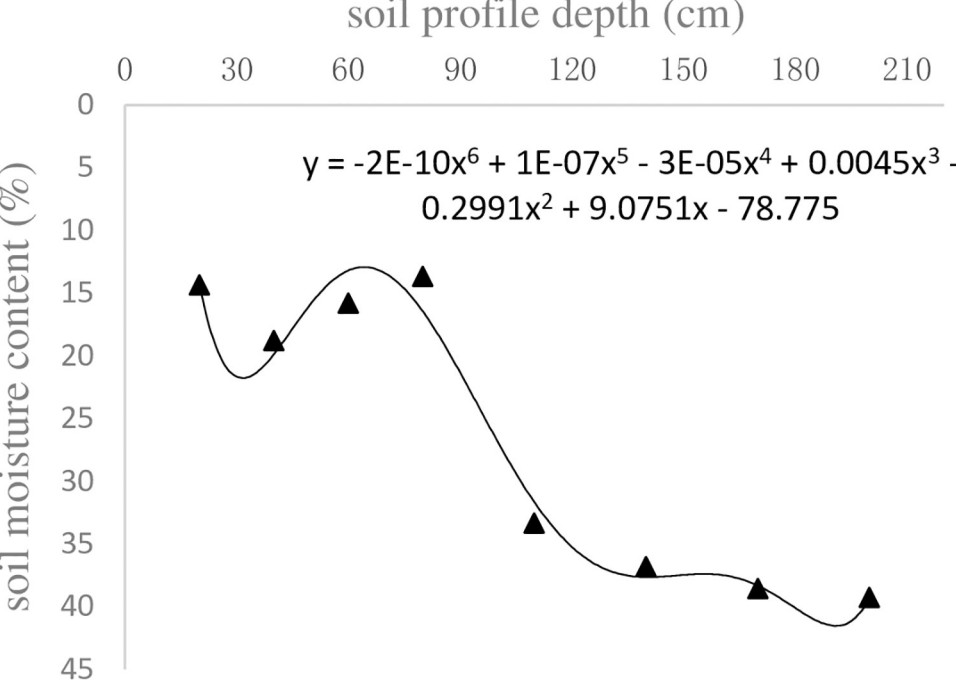

**Fig 3. Relationship between initial moisture content and depth.**

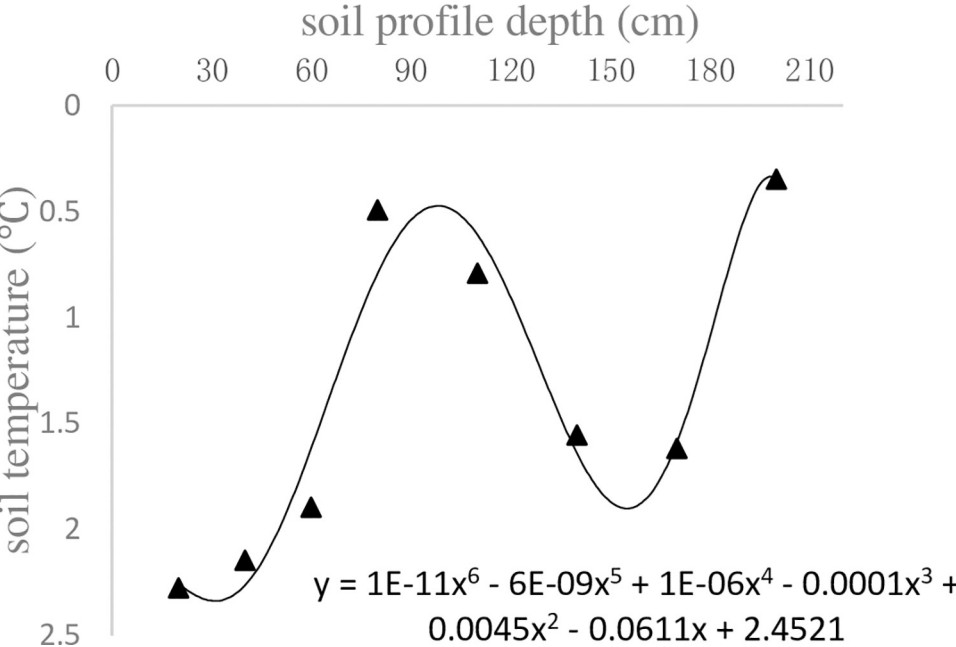

**Fig 4. Initial temperature and depth.**

For the lower boundary of soil moisture content, because the underground water level was deeper, it had low influence on the upper soil moisture content; therefore, the lower boundary of soil moisture content was taken as the first kind of boundary, i.e., the lower boundary of soil moisture content was already known.

**(2) Soil temperature boundary condition.** The upper boundary for soil temperature directly adopted the automatic meteorological station's measured value, while the lower boundary for soil temperature used the measured value.

Because the winter temperature in Xinjiang Province was not particularly low, according to past experience, the deepest frozen depth was approximately 120–150 cm; therefore, at the lower edge (200 cm), the soil was not frozen and the temperature change was not large. By contrast, during the season of melting, the soil temperature exhibited a slight decline.

## C. Seasonal frozen soil moisture characteristic parameters

**(1) Soil freezing characteristic curve.** The soil freezing and thawing process is a very complicated process involving material change and energy transfer, and it has an impact on soil and external environmental conditions. Based on the relationship between the actual measurements of non-frozen moisture content and soil temperature [33], we can obtain their mathematic relationship. The soil freezing characteristic curve can be represented as:

$$v_u = 5E - 31T^{12.819} \tag{34}$$

Where $v_u$ was the non-frozen moisture content in the soil. $T$ was the soil temperature.

**(2) Unsaturated soil moisture movement parameter.** Unsaturated hydraulic conductivity is related to saturated hydraulic conductivity (K) and is a very important parameter in the soil water flow equation. The value of this parameter was equal to soil moisture transport flux, from which an unsaturated soil unit gradient results. Unsaturated hydraulic conductivity along with changes in soil matric potential or moisture content was more difficult to derive by

theory, and an empirical formula was used to calculate it. The commonly used formula was:

$$K = as^{-m} = K_s \left(\frac{v}{v_s}\right)^m \tag{35}$$

Where $a$ and $m$ were the empirical coefficients.

Water capacity ($C$) was the change in soil moisture content from which changes in unit matric potential result. Water capacity is also related to soil moisture and soil matric potential (unit is $pa$) and the important factor of soil moisture migration; it was defined as:

$$C = \frac{dv}{d\psi_m} \tag{36}$$

Where $\psi$ was related to soil suction $s$, expressed as $s = -\psi_m$. To facilitate analysis, an empirical formula was used:

$$s = av^b = a(v/v_s)^b \tag{37}$$

Where $v_s$ was the saturated moisture content, and $a$ and $b$ were the empirical coefficients. Thus, $C$ can also be written as:

$$C = -\frac{dv}{ds} \tag{38}$$

## Material and methods

Based on local meteorological data, the physical experimental observation parameters were as follows: atmospheric pressure, light intensity, $CO_2$ concentration, wind speed, wind direction, air temperature, relative humidity, precipitation, evaporation, soil temperature and humidity, frozen soil depth. The time step was set as 20 minutes. The sandy loam soil depth was 30 cm, 60 cm, and 90 cm. In the process of solving the model, according to the actual situation, two classes of soil texture with different step sizes were mainly used to solve the model. The changes in the clay soil characteristic parameter were not large. We appropriately increased the step size, but the changes in the sandy loam characteristic parameter were larger; thus, the measurement step size was reduced. The main soil parameters are shown in Table 1. The soil temperature and moisture were measured by a soil freezing and thawing sensor that utilized an annular probe, which was developed by our own laboratory, as shown in Fig 5. Data numerical calculations used MATLAB to solve the model. Water vapor transport was tested and calculated aimed at analyzing the different soil textures during freezing and thawing period.

The main experiment contents included the following: soil moisture observations during the freezing and thawing period, soil temperature change observations, and meteorological conditions observations.

**Table 1. Clay and sandy loam soil parameters.**

| Soil texture | Soil particle diameter, mass percentage(%) | | | Rising height of capillary water(cm) | Water supply degree /(m₃m⁻³) |
|---|---|---|---|---|---|
| | Clay($<$0.0039mm) | Powder(0.0039~0.02mm) | Sand($>$0.02mm) | | |
| Clay | 81.6 | 9.5 | 8.9 | 195 | 0.005 |
| Sandy loam | 19.6 | 24.3 | 56.1 | 185 | 0.08 |

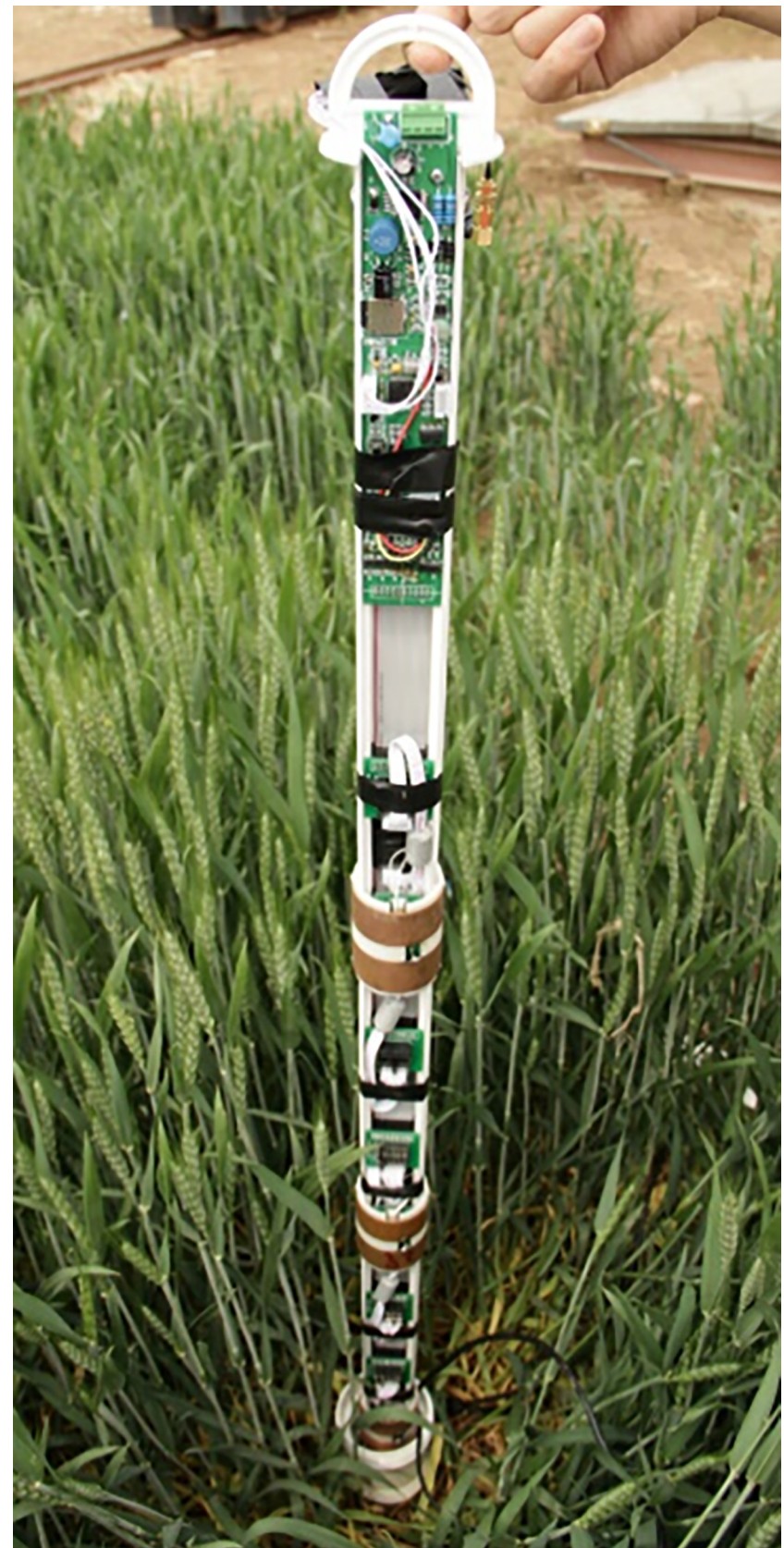

**Fig 5. Field ring probe of soil freezing and thawing sensor [41].**

## A. Soil freezing and thawing sensor structure

In this paper, the soil profile sensor structure was composed of a main board, a moisture sensor, a temperature sensor, a connecting cable, a PVC bracket and a PVC sleeve, as shown in Fig 6. A copper detector was layered (interval 10 cm, adjustable) and sheathed on the PVC bracket with a cylindrical structure. A water content detecting circuit was placed inside, and two brass electrodes were embedded on the outside to receive and transmit electromagnetic waves. The moisture probe determines the number and position according to actual needs (at most 8 sensors can be installed); the temperature sensor main board and water sensor circuit were connected by connectors, which were plugged into the connecting cable. The sensor main board collected data and controlled each layer's sensor by the connecting cables. To reduce the overall sensor power and avoid electromagnetic interference between the moisture sensors, the sensor board adopted a time-sharing power supply mode to supply the moisture sensor and temperature acquisition board [41].

Before using the sensor, it was necessary to use the specified tool to bury a PVC tube at the test point. During the test, according to the actual demand, the moisture probe position and number of probes were adjusted. The sensor was inserted into the PVC tube and was connected to a power supply and a data cable that was tightly covered. Then, this sensor was able to collected online, real-time measurements of soil moisture and temperature at various depths.

**(1) Electric field distribution of annular probe.**   The ring probe model was established by using HFSS electromagnetic field simulation software. The frequency was set to 100 MHz and the lumped port excitation mode was selected; the probe diameter was 2.5 cm, and its widths were 2.0 cm, 2.5 cm, 3.0 cm and 3.5 cm. The dielectric constant around the filling medium was set to 21 (corresponding to the soil volume water content of 36%). The dielectric constant of the PVC pipe installed with a copper ring bracket was 4. The medium in the pipe was set to air. The dielectric constant was set to 1; the copper ring electrode was set as the ideal boundary of the electric field; the diameter was set to 12 cm; the height was set to 13 cm; and the cylinder represented radiation boundary conditions.

It can be seen from Fig 7 that the four structures of the water content probe were mainly distributed evenly between these probes, and the surrounding electric field was compact without a separation phenomenon. The different widths of the annular probe mainly affected the longitudinal and horizontal ranges of the monitored electric field distribution. The electric field strength was 104.9 V/m in the pale gray area, and the scope of the probe's longitudinal electric field strength color tended towards the light gray area, which increased as the width of the copper ring increased to 9.5 cm, 10 cm, 10.5 cm and 11 cm. In contrast, the range of the probe's electric field strength reaching the light gray region, which decreased with increasing copper ring width, decreased to 10.5 cm, 10 cm, 9.5 cm and 9 cm. From the above phenomenon, it can be ascertained that these four ring electrode widths were all suitable for use in the sensor detection probe. In a practical application, the appropriate annular electrode width should be selected by considering the horizontal and vertical detection area. The conventional separation interval in agricultural applications is 10 cm; the copper ring with a width of 2.5 cm was chosen as the sensor electrode in this paper.

The PVC tube had attenuation effects on the electric field strength. The copper ring internal electric field intensity was higher than outside PVC electric field strength. The copper ring internal moisture test board was considered to influence the electric field distribution; in this paper, a metal shield was used on the moisture test mainboard.

**(2) Performance test of sensor dynamic response.**   1) The sensor's dynamic response performance mainly reflected the time required for the sensor to respond when the water content changed in the sensor detected area. This experiment analyzed the sensor's moisture detection

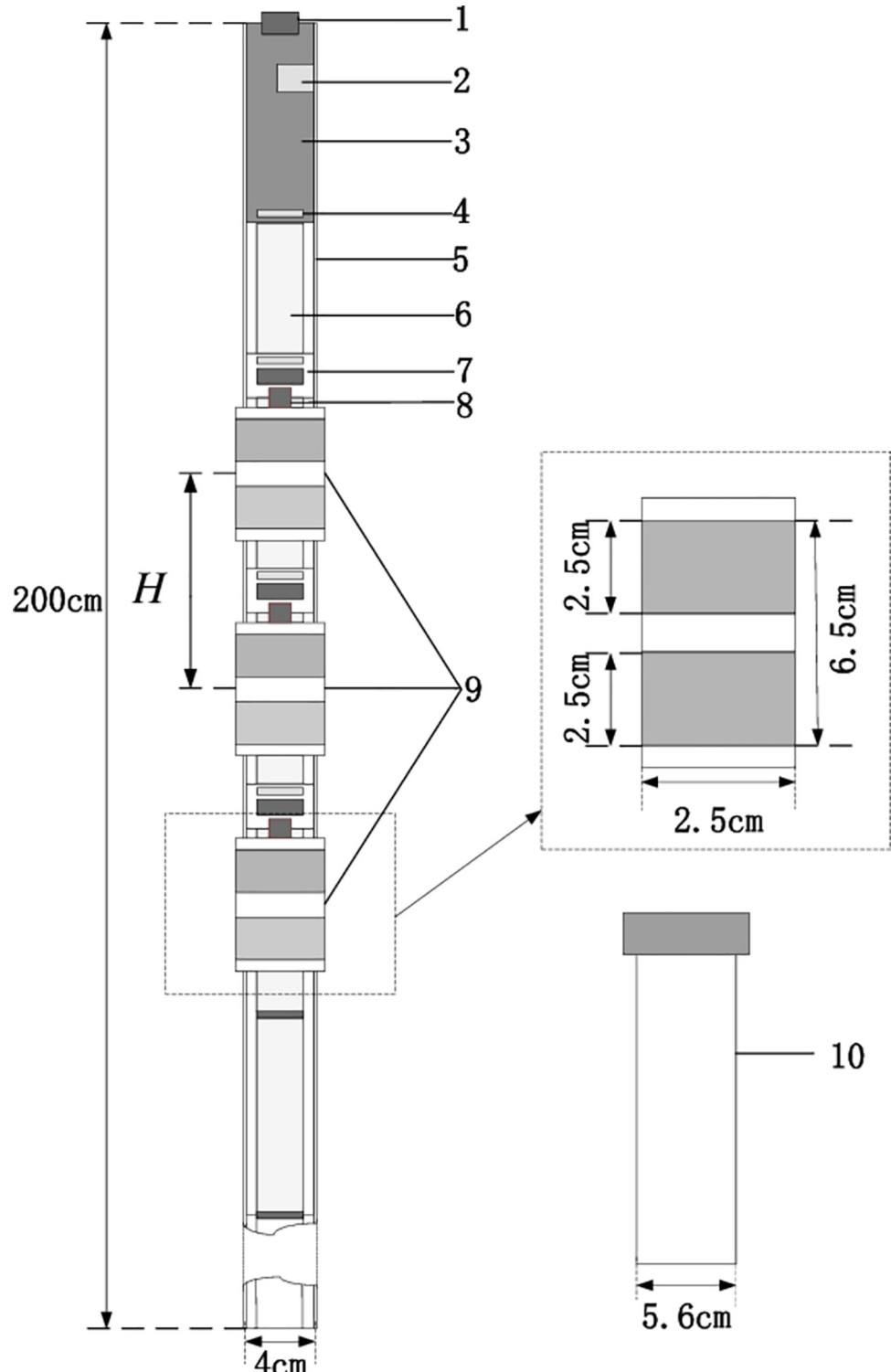

**Fig 6. Schematic of soil freezing and thawing sensor** [41]. H. Spacing of sensor probe (The shortest distance was 10cm) 1. Communication interface of power supply and RS485 2. SD card storage module 3. Sensor collecting mainboard 4. Bottomed cable connector 5. PVC support frame 6. Connecting cables 7. Temperature sensor board 8. Moisture and temperature mainboard connector 9. Copper detection probe 10.PVC tube.

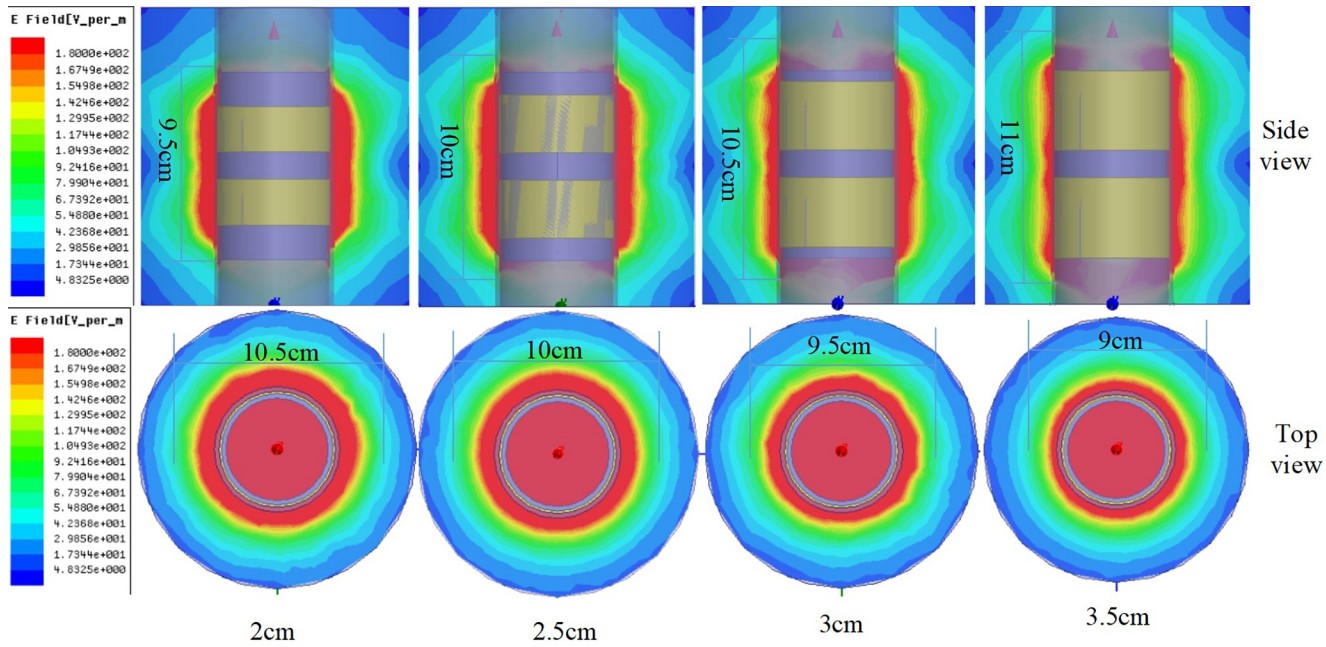

**Fig 7. Simulation of electrical field distribution of a ring probe.**

capabilities, and the PVC test tube was placed in water. An oscilloscope was used to detect the time required for the sensor to switch from being just energized to producing a stable output. The result is shown in Fig 8. $\Delta X$ was the time difference between sensors from energization to the point at which a stable output was maintained. The dynamic response time was 1.28 ms.

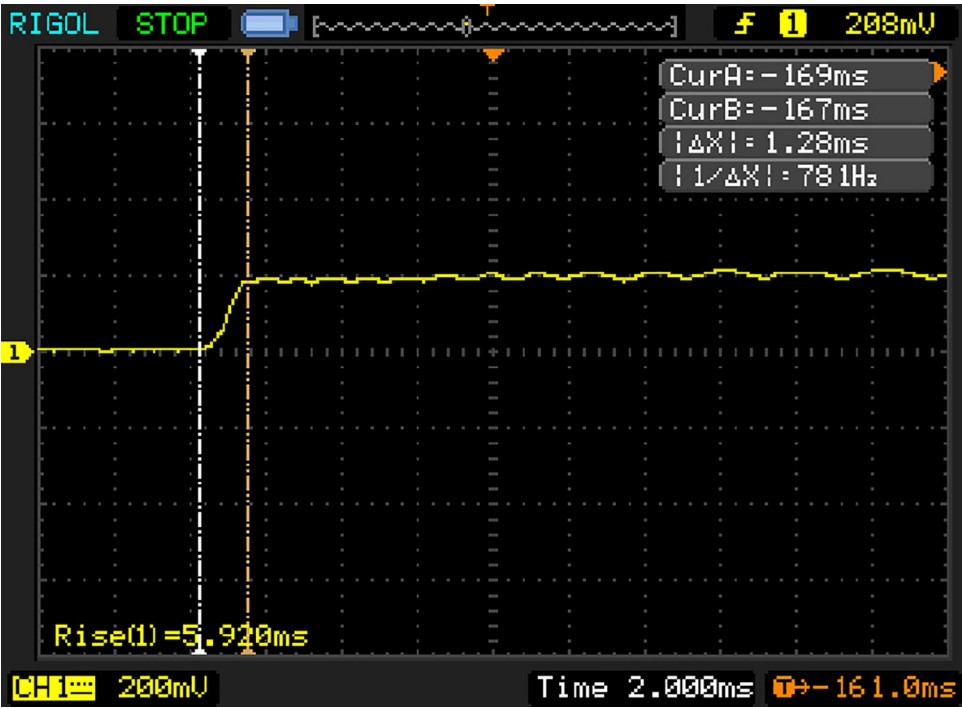

**Fig 8. The dynamic response of soil moisture sensor.**

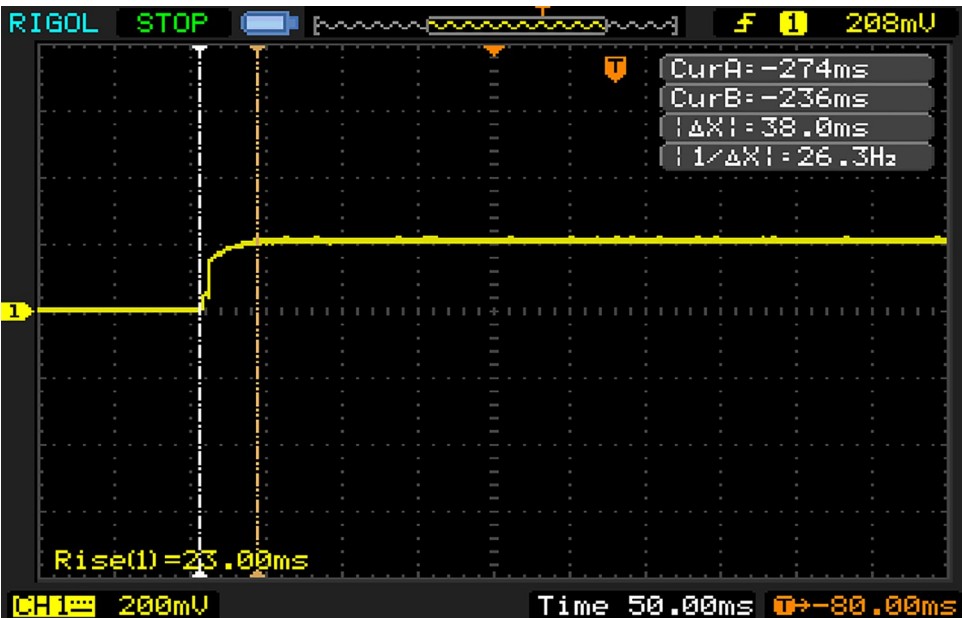

**Fig 9. The dynamic response of temperature sensor.**

2) The temperature detecting element adopted an armored platinum resistance, and the temperature measuring circuit response time principle was similar to that above. The dynamic response time was 38.0 ms after each test, the result is shown in Fig 9.

## B. Soil moisture observation

The soil moisture observation experiment mainly included initial soil moisture, total water content and unfrozen water content observations. The sensor was used to measure the initial moisture and total water content when the freeze had not yet started. A homemade freeze thaw sensor was adopted to measure the unfrozen water content during the freezing period. The measurement method used was that of the soil profile determination; the actual schematic layout is shown in Fig 10.

As seen from Fig 10, the freezing and thawing data were embedded with 8 pipes. The tube was a high strength PVC plastic pipe, the length was 2000 mm, the inner diameter was 58 mm, and the outer diameter was 60.3 mm. The test tube was consolidated in August 2017 before the soil was unfrozen, and testing began after a few months of stabilization, the sensor layout was shown in Fig 11.

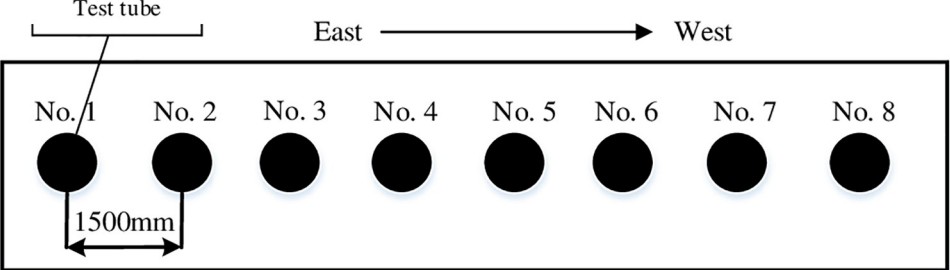

**Fig 10. Schematic layout of test tube (mm).**

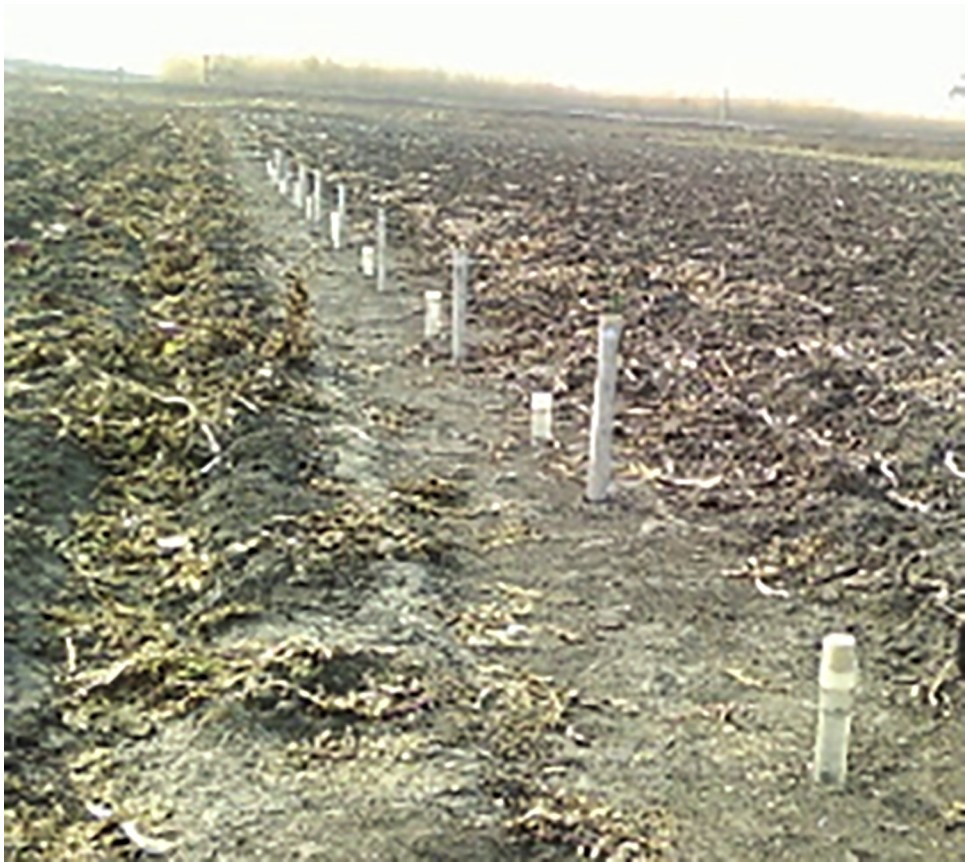

**Fig 11. Sensor layout.**

### C. Soil temperature observation

A DS18B20 digital temperature sensor was used to collect soil profile temperature observations. During the experiment, temperature sensors and soil moisture sensors were placed in a vertical arrangement as shown in Fig 12 in the test area.

## Experiment result & verification

First, the experimental equipment was installed. The sensor equipment was installed on Nov. 20, 2017, and the first observation data were collected on Jan. 20, 2018. The soil hydrothermal conditions were observed; the soil was not completely frozen at this time. Simulation and experimental results are shown in Fig 13.

As seen in Fig 13, the soil moisture at a depth of approximately 80 cm began to freeze because the latitude was high, air temperature was low, upper layer of soil was influenced considerably by the external environment, and temperature changed rapidly with the depth of the soil. The soil was uniform in texture, located in a sandy soil zone, and exhibited slow change in the lower-layer moisture content; this transformation agreed with the actual situation. In addition, the simulation result from numerical analysis was consistent with the trend in variation measured in the experiment, and the error associated with the surface soil was smaller than that associated with deeper soil, which may be due to surface evaporation or the low accuracy of the parameter values for the lower soil thermal characteristics. The errors between measured results and simulation results are listed in Table 2 as well as discussed below.

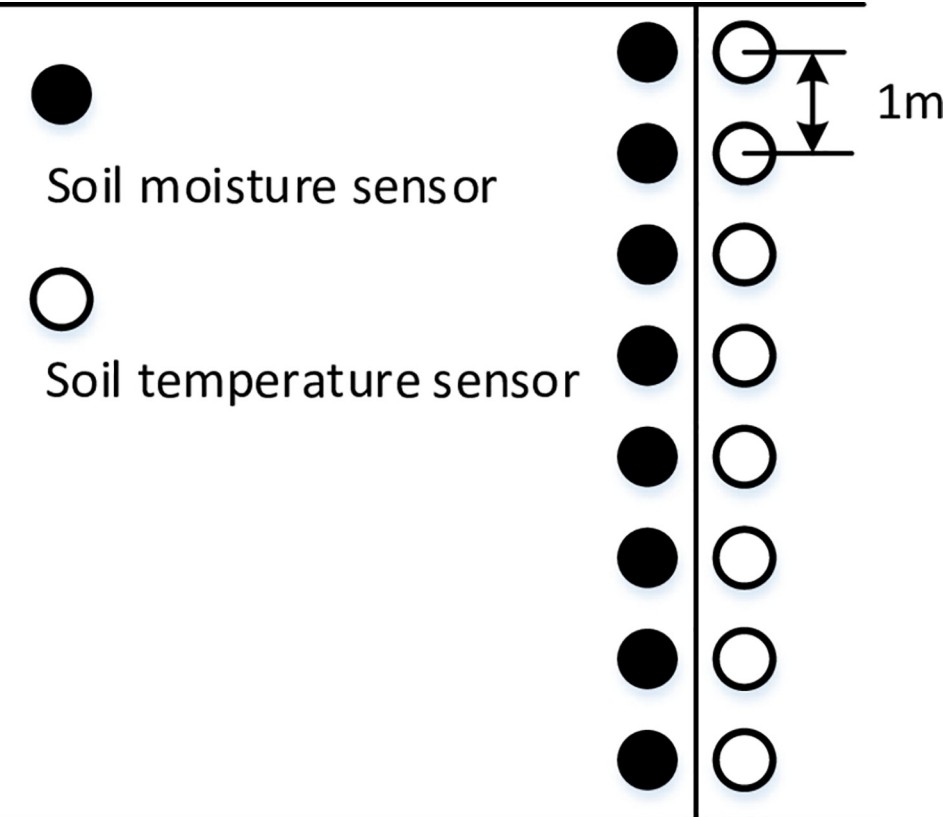

**Fig 12. Sensors buried sectional.**

As seen in Table 2, the largest relative error was 4.36, minimum error was 0.98, and average error was 2.515. The simulation result of the period of soil freezing was reliable. The numerical simulation results were generally consistent with the measured data, but the error associated with the deep-layer soil was large because the subsoil was sandy soil, and the water flow was larger. In the subsoil from a depth of 100 cm to the surface, moisture change was larger. Because day and night exhibited large temperature differences at high latitudes, the climatic conditions caused air and surface temperature transmission; thus, the simulation result was similar to the real values. Next, the soil temperature experiment was performed and the result is shown in Fig 14.

The soil temperature simulation result was similar to the result for moisture content (Jan. 20, 2018). The result had a certain amount of error, but the change trend was the same. As seen in Fig 14, the frozen soil frost at a depth of approximately 80 cm began to change, and the surface temperature was slightly higher, because the freezing period did not have snow cover, the combined ultraviolet irradiation had high intensity, the temperature was higher than in the previous year in this area, and the soil temperature was approximately 1.5°C. In addition, the upper soil temperature changed rapidly with depth, which suggested that the lower sand was experiencing the freezing period. The freezing front surface from water to ice was approximately 200cm underground (considering the boundary condition, the sensors could not be deployed deeper). The errors between the measured result and the simulation result are listed in Table 3 as well as discussed below.

Because the temperature sensor's sensitivity was good, the errors between numerical results and measured values were large; specifically, the largest relative error was 14.31,

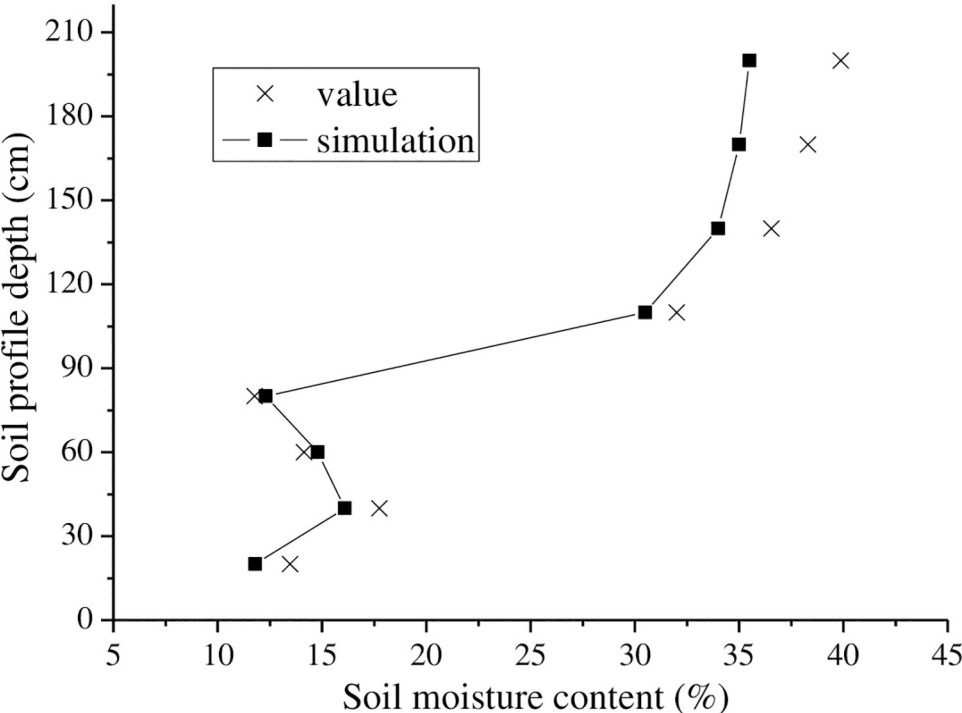

**Fig 13. Measured value compared with simulated value of soil moisture content (2018-01-20).**

the minimum relative error was 0.67, and the average error was 6.67. These errors were within the permissible error range, indicating that the simulation results were generally reliable.

## Results and discussions

### A. Qualitative analysis of soil water diffusion and evaporation

In this paper, the influence factors of hydrothermal coupling model included two aspects: water vapor diffusion and evaporation [34]. In water migration conveying process, in the process of the migration of water content, movement direction, and transport intensity will change with time, on which effected the change of the frozen soil moisture and temperature [35, 36]. For the air column in one-dimension vertical in the frozen soil, upper bound was taken as top gas convection column, lower bound was soil bottom surface [37, 38]. According to water balance principle, the air column of atmospheric water balance can be built as [39]:

$$(W_1 + E_i) - (W_2 + P_i) = \Delta W \tag{39}$$

$W_1$ was water vapor content which flowing into gas column, $W_2$ was the water vapor content which outflowing gas column, $E_i$ was evaporative emission, $P_i$ was precipitation, $\Delta W$ was

**Table 2. Relative error of measured values for water content.**

| Dept (cm) | 20 | 40 | 60 | 80 |
|---|---|---|---|---|
| Relative error (%) | 3.26 | 2.05 | 2.53 | 0.98 |
| Dept (cm) | 110 | 140 | 170 | 200 |
| Relative error (%) | 3.87 | 3.91 | 4.19 | 4.36 |

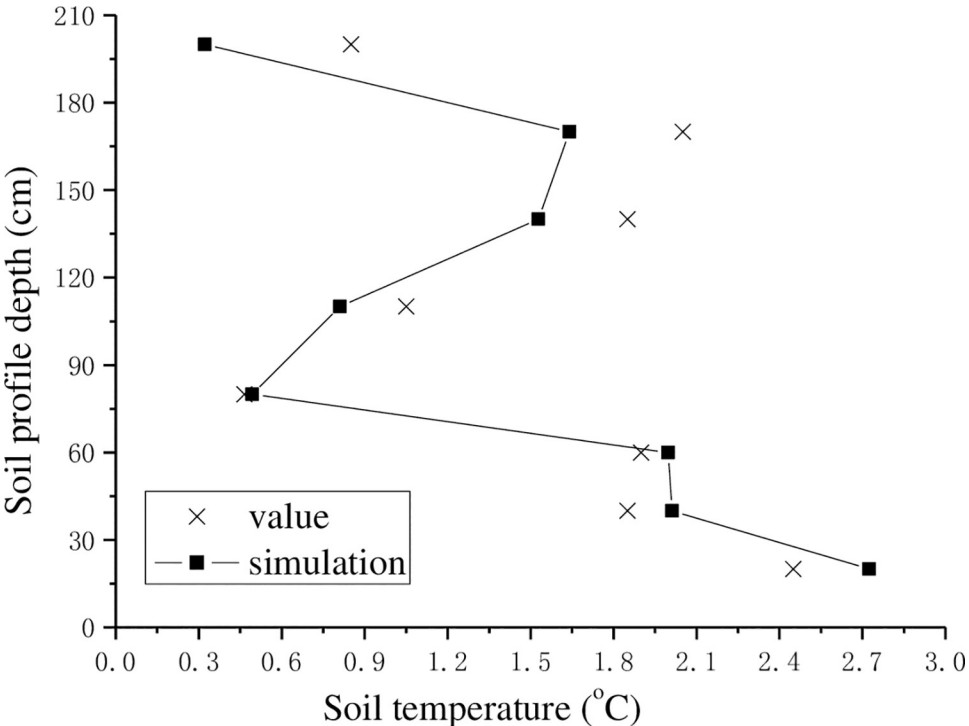

**Fig 14. Measured value compared with simulated value of soil temperature (2018-01-20).**

water vapor variable of gas column, for a long time observation, $\Delta W \rightarrow 0$, so one-dimension vertical frozen soil column water movement can be represented as:

$$W_A = W_1 - W_2 = P_i - E_i \tag{40}$$

$W_A$ was water vapor content. Because the evaporation rate was much less than the delivery value of water vapor, the precipitation of the region (snowfall) determined water vapor amount in frozen soil column [40]. At the same time, in the process of water vapor transport, soil also accompanied momentum and heat transfer, it affected soil temperature, pressure and other model parameters.

According to the observed data from the test station, the information of the change of the soil moisture was recorded from December 2017 to March 2018. According to the simulation results, surface soil evaporation latent heat evaporate emission was $E_i$, $P_i$ can be acquired by rain gauge or local weather information. According to Eq (40), the frozen soil moisture transfer simulation results were qualitatively analyzed in the freezing and thawing period in the test area, the results were shown in Fig 15.

The qualitative results of one-dimensional vertical frozen soil columns under different soil texture and buried depth can be seen from Fig 15. With the depth of the soil with different

**Table 3. Relative error of measured temperature values for water content.**

| Dept (cm) | 20 | 40 | 60 | 80 |
|---|---|---|---|---|
| Relative error (%) | 7.11 | 5.14 | 2.91 | 0.67 |
| Dept (cm) | 110 | 140 | 170 | 200 |
| Relative error (%) | 4.36 | 6.32 | 14.31 | 12.53 |

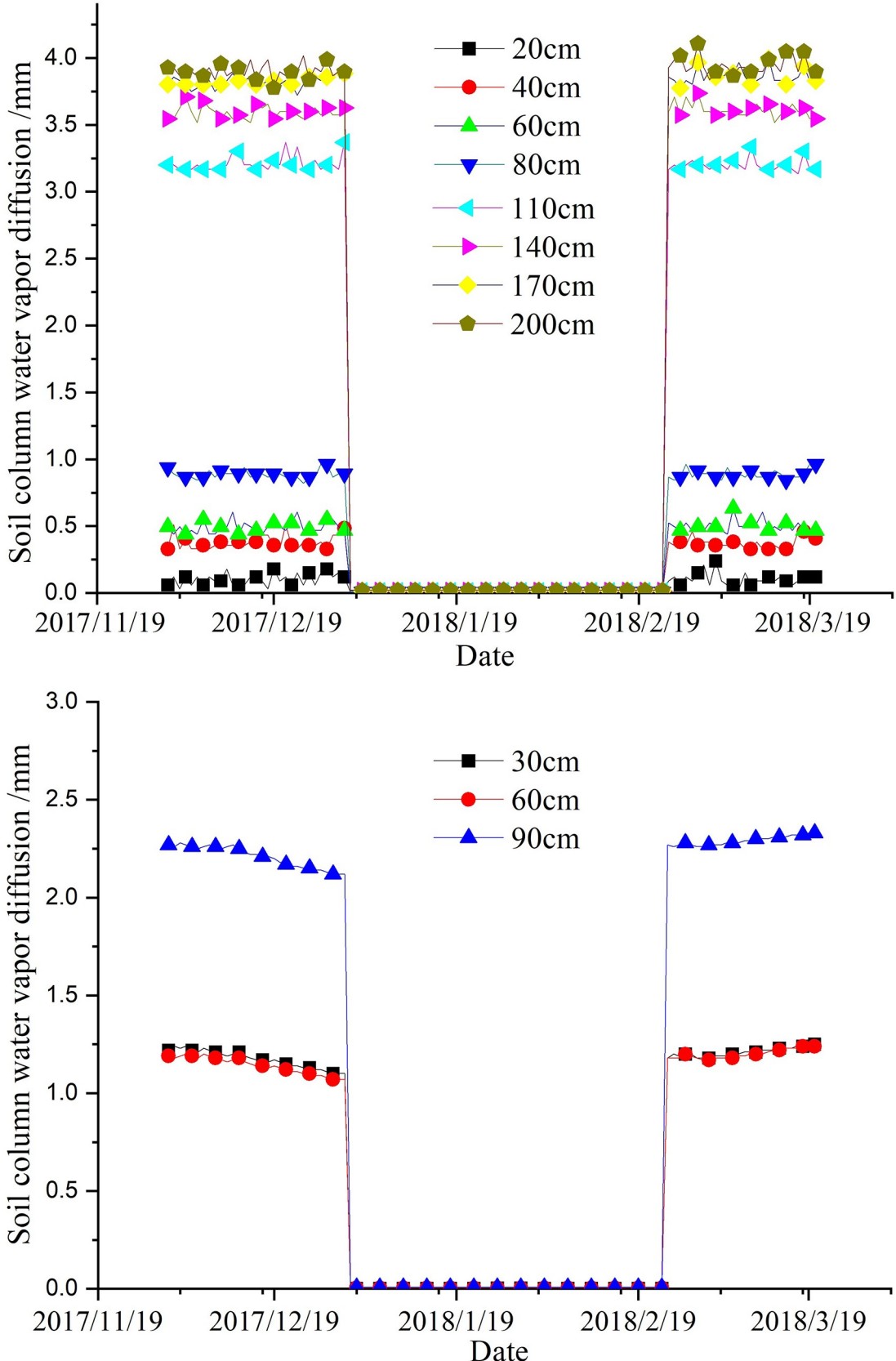

**Fig 15. A qualitative analysis of vertical frozen soil pillar water vapor diffusion (Zhang Chaoyi et al. 2018).** (a) clay. (b) Sandy loam soil.

texture going deep, the surface soil evaporation was small, deep soil moisture diffusion phenomenon was more obvious. From December 2017 to March 2018, during a complete freezing and thawing period, the deepest phreatic water level's evaporation was highest, clay and loamy sand average evaporation were respectively 1.93mm/day and 1.68mm/day. Because the saturated hydraulic conductivity was different, the phreatic soil depth evaporation and moisture diffusion were large. In clay surface soil and 1m below bottom soil, water evaporation and diffusion content were about 3.51mm/day, while sandy soil water evaporation and diffusion were obvious changed in soil depth (0.6~0.9m), the average difference was about 0.98mm/day. Because every year's January marked the northern winter, frozen soil was relatively in a stable stage, the freezing depths prevented the water contact of the subsoil and outside atmosphere, of which cut soil column from frozen depth, so above frozen depth the soil can take water vapor evaporation, diffusion and external atmosphere etc. As a result, the water vapor transport amount was small, the lower subsoil almost had no good interaction with environment, it only interacted with deeper groundwater for water vapor transport process, Eq(40)'s $E_i$ was a small value, caused soil column moisture evaporation larger.

## B. Effect of soil texture on water vapor diffusion

It was difference between soil particle's interspace, which made soil water potential, unsaturated hydraulic conductivity and other soil parameters characteristics differ. In one-dimensional vertical frozen soil column model, water movement and evaporation were closely related to hydraulic effect, this waterpower affected water and heat coupling. In Fig 15, under the same depth (60cm) and near the layer depths, contrastive analysis showed that sandy loam soil moisture evaporation was faster, air convection was more, moisture diffusion was in large quantity. With the increasing depth, the increment of the diffusion of the water vapor of the sandy loam was more obvious. Because clay has good water retention, the diffusion of the moisture of the freezing and thawing soil was generally lower than sandy loam. Above frozen soil layer (100cm), moisture diffusion amount was analyzed, clay soil moisture evaporation average amount was about 0.51mm/day, sandy loam soil moisture evaporation average amount was about 1.24mm/day, which was 241.99% times than clay. It is relatively that the difference of soil texture influencing on the moisture migration of the frozen layer.

## Conclusion

By comparing numerical simulation results and experimentally measured data, this paper proposed a new hydrothermal coupling differential unit cell model that is based on a sphere. The seasonal frozen soil hydrothermal coupling equations were successfully derived, and the corresponding calculation method was applied to simulate the soil-freezing process. Compared with the traditional parallelepiped basic differential unit model, the proposed model has increased computational complexity. According to the experiment results, this model had certain errors, characteristic parameters such as the automatically adjusted step length (time and space) need to be improved, and its accuracy should be further improved. The conclusions are as follows:

1. In high latitudes of northern China, the soil surface is affected by cold climate in winter, so the freezing and thawing effect is obvious. From the experiment data, November to January is the winter of northern China, the depth of permafrost changes with temperature. The freezing depth basically changes at 30cm-200cm in soil surface profile. The water vapor, moisture, and temperature changes and transitions are also carried out, in addition, from the perspective of soil quality, with the increasing of soil depth, the water and heat migration becomes more

and more intense, this is because the soil is sandy and the moisture flow is large. The water changing is the most dramatic about 100cm below the surface, which causes the drastic interaction of air and surface soil's temperature and moisture. Due to the large temperature difference between day and night, the error between the numerical simulation calculation and actual measured data is within a reasonable range.

2. For the temperature changes, the proposed sphere model is also simulated in this region, and the results are consistent with the above conclusion for different soil and frozen depth. From the results, the temperature variation located at the soil frozen depth of 80 cm shows an inflection point, because there is no snow cover during the freezing period in winter, the high intensity of local ultraviolet radiation makes the freezing depth shallow, the soil temperature maintains at about 1.5°C. Moreover, from the perspective of soil quality, the temperature of soil lower layer decreases with increasing depth, which indicates that the lower sandy soil is in the freezing period, and the freezing front of the transformation from water to ice is about 2.5 meters underground. This result shows that the freezing depth of sandy soil is deep, the change of soil temperature is obvious, and the conclusion has scientific significance for soil moisture conservation and irrigation.

In addition, experiments were also carried out late in the freezing period. As this period was accompanied by human intervention (irrigation), some rain, and snow interference, the characteristic parameters changed considerably, the final matching curve differed greatly from the measured values, but their basic trends were consistent.

## Supporting information

**S1 File. Data in brief.**
(RAR)

**S2 File. Research data.**
(RAR)

**S3 File. Supplementary material.**
(RAR)

## Acknowledgments

The authors would like to thank the anonymous reviewers for their constructive comments.

## Author Contributions

**Conceptualization:** Chaoyi Zhang, Feng Chen.

**Data curation:** Lei Sun.

**Formal analysis:** Chaoyi Zhang, Zhangchao Ma, Yan Yao.

**Investigation:** Feng Chen, Lei Sun.

**Methodology:** Chaoyi Zhang, Feng Chen.

**Project administration:** Chaoyi Zhang.

**Supervision:** Yan Yao.

**Writing – original draft:** Chaoyi Zhang.

**Writing – review & editing:** Chaoyi Zhang, Lei Sun, Zhangchao Ma.

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
