## [Decision Letter · Decision Letter 0]

18 Jun 2021

PONE-D-21-16032

A new seasonal frozen soil water-thermal coupled migration model and its numerical simulation

PLOS ONE

Dear Dr. Sun,

Thank you for submitting your manuscript to PLOS ONE. After careful consideration, we feel that it has merit but does not fully meet PLOS ONE’s publication criteria as it currently stands. Therefore, we invite you to submit a revised version of the manuscript that addresses the points raised during the review process.

We look forward to receiving your revised manuscript.

Kind regards,

Jianguo Wang, PhD

Academic Editor

PLOS ONE

Journal Requirements:

Additional Editor Comments (if provided):

Reviewers' comments:

Reviewer's Responses to Questions

**Comments to the Author**

1. Is the manuscript technically sound, and do the data support the conclusions?

Reviewer #1: Yes

Reviewer #2: Yes

2. Has the statistical analysis been performed appropriately and rigorously? 

Reviewer #1: Yes

Reviewer #2: Yes

3. Have the authors made all data underlying the findings in their manuscript fully available?

Reviewer #1: Yes

Reviewer #2: Yes

4. Is the manuscript presented in an intelligible fashion and written in standard English?

Reviewer #1: No

Reviewer #2: Yes

5. Review Comments to the Author

Reviewer #1: This paper presents a mathematical model that utilizes a spherical differential unit cell as a spatial infinitesimal differential unit cell model for describing seasonal soil freezing and thawing. The test data of a seasonally frozen soil in the Shihezi of Xinjiang Province was used to verify the model. The results showed that the model is suitable. There are some suggestions for the manuscript:

(1) Some water-thermal coupled models have proposed in a rectangular coordinate system, the authors just convert the models to the spherical coordinate system rather than derive the model in a the spherical coordinate system.

(2) The model in a spherical coordinate system is inconvenient to simulate the water and heat transfer of a soil which shape is not a sphere, so what are the advantages of the model?

(3) In Fig. 3, the lines of the values of temperature and soil moisture are incorrect; and the authors need to review the data.

(4) The conclusions need to be more specific.

(5) The authors are encouraged to improve the language in this manuscript.

Reviewer #2: It is true that seasonally frozen soils at high latitudes are a topic of major interest in soil physics. This paper presents a mathematical model that utilizes a spherical differential unit cell as a spatial infinitesimal differential unit cell model for describing seasonal soil freezing and thawing. Using this model, the basic equations for water and heat flow through frozen soils are directly derived, and these derived equations are combined with a contact equation to create a frozen soil water-thermal coupled migration model. It can provide a reference for the soil freezing and thawing state. The reviewer would like to propose the following comments based on the contents of the manuscript.

1. The Introduction is not good. A good introduction should include current research progress and deficiencies. In this paper, the authors describe many details of current research. Third power computation methods should not be used for the derivation of actual frozen soil thermal coupled water transport equations. The highlights and the scientific contributions of this manuscript need to be emphasized.

2. In Mathematical model, the mathematical calculation model of single-hole frozen soil column was proposed. The heat conduction equation, boundary conditions and Initial conditions were presented. However, the biggest problem is the phase transition of frozen soil. How do you consider the phase transition? This is the most important characteristic of frozen soil.

3. This paper proposed a new hydrothermal coupling differential unit cell model that is based on a sphere. Compared with the traditional parallelepiped basic differential unit model, the proposed model has increased computational complexity. What are the advantages of this study?

4. Based on local meteorological data, the physical experimental was carried out. But the detailed engineering geological conditions, hydrogeological conditions and construction freezing process are not introduced. This is the key. The reliability and discreteness of the result are doubtful.

6. PLOS authors have the option to publish the peer review history of their article (what does this mean?). If published, this will include your full peer review and any attached files.

Reviewer #1: No

Reviewer #2: No

---

## [Author Response · Author response to Decision Letter 0]

24 Jul 2021

Thank the reviewers for reviewing this article, and put forward some very constructive opinions. According to these opinions, the authors carefully revised paper, including re-analysis mathematical model, re-testing, re-writing the introduction, re-translating the whole manuscript, etc., and answered the reviewers' questions one by one, the details are as follows:

Reviewer #1: This paper presents a mathematical model that utilizes a spherical differential unit cell as a spatial infinitesimal differential unit cell model for describing seasonal soil freezing and thawing. The test data of a seasonally frozen soil in the Shihezi of Xinjiang Province was used to verify the model. The results showed that the model is suitable. There are some suggestions for the manuscript:

(1) Some water-thermal coupled models have proposed in a rectangular coordinate system, the authors just convert the models to the spherical coordinate system rather than derive the model in a the spherical coordinate system.

Answer: Yes, this is an important innovation of this paper. In the rectangular coordinate system, the author replaced the previous cube model with the spherical model. This is because in the soil’s spatial distribution, the spherical model is more accord with the general movement law of water and water vapor in the actual situation, however, there will be errors in the edges and corners of the cube for the random flow of water. Therefore, the author uses the spherical model to deduce the water heat coupling equation. The choice of coordinate system is rectangular coordinate system, because the radius and angle of spherical coordinate system can be converted into the spatial correspondence of rectangular coordinate system, so this paper focuses on the minimum spatial physical unit of soil water movement, which is a spherical structure model.

The above content is added to the corresponding position of the last paragraph in the introduction section of the paper, and marked it.

(2) The model in a spherical coordinate system is inconvenient to simulate the water and heat transfer of a soil which shape is not a sphere, so what are the advantages of the model?

Answer：Based on the cube model introduced in the previous literature, this paper innovatively proposes the physical process of water movement under sphere model. The biggest advantage of this model is that the particle can do any trajectory movement in 360 degree free space. As a water molecule, it can enter and exit the spherical structure designed in 360 degree free space.

The former cube micro units can also analyze and study the migration of water molecules. However, the cube has "edge" and "angle". When water molecules pass through the edge and angle, they are not very good at calculating its motion. However, the sphere structure has no "edge" and "angle", which fully proves that water can be transported in any direction, which is more in line with the actual situation, Therefore, this paper proposes such a new spherical micro unit for induction, which is the basis of analyzing the law of water movement.

The above content is added to the corresponding position of the first paragraph in the part of “Mathematical model” � “A. seasonal frozen soil water movement.”

(3) In Fig. 3, the lines of the values of temperature and soil moisture are incorrect; and the authors need to review the data.

Answer: Yes, we reviewed the previous data of initial soil moisture content and corrected the soil moisture and temperature curves in Figure 3 and Figure 4. Please refer to the revised version of the manuscript.

(4) The conclusions need to be more specific.

Answer: we reorganized the conclusions. From the experimental results and numerical fitting, we elaborated and analyzed the new hydro thermal coupling model proposed in this paper. Please refer to the revised manuscript. 

As following:

“……The conclusions are as follows:

1. In the high latitude area of northern China, winter is affected by cold climate, and the effect of freezing and thawing on soil surface is obvious, and different soil properties are also affected differently. From the experiment data, November to January is the winter of northern China, and the depth of frozen soil changes with the temperature. The frozen depth basically changes at 30cm-200cm of the soil surface profile, and the water vapor, water content, and soil moisture content are different The change and conversion of temperature are also carried out in this region. In addition, from the perspective of soil quality, with the increase of soil depth, the water and heat migration becomes more and more intense. This is because the soil is Sandy and the water flow is large. The water changing is the most dramatic about 100cm below the surface, this causes the water of air and soil surface interact violently, due to the large temperature difference between day and night, therefore, the error between the numerical simulation results and measured data is within a reasonable range.

2. For the temperature changes, the proposed sphere model is also simulated in this region, and the results are consistent with the above conclusion for different soil and frozen depth. According to the results, there is a turning point in the temperature change at the depth of 80cm, this is because in winter, there is no snow cover in the freezing period, and the local ultraviolet radiation intensity is high, which makes the freezing depth shallow, and the soil temperature is moderate and maintained at about 1.5℃. In addition, from the perspective of soil quality, the temperature of soil lower layer decreases with increasing of depth, which indicates that the lower layer of sandy soil is in the freezing period, and the freezing front from water to ice is about 2.5 meters underground. This result shows that the freezing depth of sandy soil is deep, and the change of soil temperature is obvious. This conclusion has scientific significance for soil moisture conservation and irrigation.

In addition, experiments were also carried out late in the freezing period. As this period was accompanied by human intervention (irrigation), some rain, and snow interference, the characteristic parameters changed considerably, and the final matching curve differed greatly from the measured values, but their basic trends were consistent.”

(5) The authors are encouraged to improve the language in this manuscript.

Answer: We re-translated the manuscript. We asked AJE(https://www.aje.cn/) professional English translation team to help us translate this article, and translated the paper according to the language standard of official English journal.

 

Reviewer #2: It is true that seasonally frozen soils at high latitudes are a topic of major interest in soil physics. This paper presents a mathematical model that utilizes a spherical differential unit cell as a spatial infinitesimal differential unit cell model for describing seasonal soil freezing and thawing. Using this model, the basic equations for water and heat flow through frozen soils are directly derived, and these derived equations are combined with a contact equation to create a frozen soil water-thermal coupled migration model. It can provide a reference for the soil freezing and thawing state. The reviewer would like to propose the following comments based on the contents of the manuscript.

1. The Introduction is not good. A good introduction should include current research progress and deficiencies. In this paper, the authors describe many details of current research. Third power computation methods should not be used for the derivation of actual frozen soil thermal coupled water transport equations. The highlights and the scientific contributions of this manuscript need to be emphasized.

Answer: We re-wrote the introduction, increased the latest progress and existing problems of frozen soil research, reduced the current research details, emphasized the scientific contribution of this paper, and updated the reference literature, making the article theme more prominent. The corresponding changes are marked in the introduction part.

As following:

“……

In the aspect of latest progress of frozen soil research, the main achievements are as follows: ① In a complete annual freeze-thaw cycle, the soil near the surface has generally experienced four stages: Summer thawing period, spring and autumn thawing freezing period, and winter freezing period. Affected by local factors, the start and end time, rate and type of freezing or ablation are different in different regions[1-3]. ② The difference of daily freeze-thaw cycle between permafrost region and seasonal frozen region is large, which is mainly reflected in the duration of daily freeze-thaw cycle[4-5]. ③ Different land surface models can well grasp the temporal and spatial changes of physical quantities in the freezing and thawing process, but they need to be parameterized according to the characteristics of land surface process[6-7]. ④ Avoiding the unstable iterative calculation, the critical temperature of freezing and thawing was determined according to the thermodynamic equilibrium equation, this can improve the unreasonable freezing and thawing parameterization scheme[8].

Literatures [9-10] considered the unfreezed water content in the cold region to improve the accuracy of the coupling simulation of heat transfer and water in frozen soil. The comprehensive algorithm and parameterization were used to calculate the thaw water content. The selected parameters of unfreezed water content were evaluated by using soil temperature, specific surface area of soil particles, soil water curve and different types of water. The results showed that the parameterization of unfreezed water content was affected by many factors, and the heating and cooling process was particularly important when calculating the unfrozen water content. Existing problems was[9-11]: there were many physicochemical parameters in these studies, which were not easy to obtain. They were difficult to be used to calculate unfrozen water content. A practical high-precision physicochemical parameter needed to be developed to couple with freezing model and land surface process model.

Literatures [12-14] considered that it was very important to determine the modeling scheme of macropore matrix interaction and osmotic water re-freezing, discussed the necessity of studying the effect of macropore flow and soil freeze-thaw interaction, the necessity of integrating these concepts into a framework of coupled water and heat transfer, put forward a conceptual model of unsaturated flow in frozen macropore soil, The model assumed that the two interacting domains (macropores and matrix) had different water and heat transfer mechanisms.

Existing problems were[12][15-16]: the detailed understanding of the mechanism of macropore flow in permafrost and how it changes with different soil thermal conditions are still uncertain in these proposed models. It is necessary to further develop the existing macropore flow description and various scale modeling methods. Testing these concepts and new modeling methods to quantify these dynamics can solve the velocity of water flow in frozen microporous soils, and study the conditions that make water flow around the frozen zone or cause water to freeze in macropores in the opposite way.

In terms of technical means, literature [17] used the interferometric synthetic aperture radar (InSAR) technology to monitor the surface of permafrost area all day and all-weather, summarized the application of D-InSAR and time series InSAR Technology in permafrost area, the research progress and future development trend of permafrost area in recent 20 years, the influencing factors of surface deformation in permafrost region were analyzed.

Existing problems were[17-19]: under the global warming, making full use of the accumulated long time series SAR data to achieve multi-directional research results of different scales and resolutions, the continuous collection and expansion of field measured data in permafrost regions will improve the existing freeze-thaw models and develop different physical parameter models, this was the trend of permafrost measurement technology in future.

In literature [20], the heat pulse (HP) method was used to determine the thermal properties and other physical properties of soil. The HP method was based on linear heat source solution of radial heat flow equation. A high pressure probe structure was proposed, the measured properties were discussed in unfrozen and frozen soil.

There were still some problems in these frozen soil observation techniques[17-23]: the current probe design limits the representative volume of soil samples. Its extreme sensitivity to needle distance leads to the lack of accuracy, accuracy and durability; The short length of thermal TDR needle affects the accuracy and accuracy of water or water flux estimation, and new and innovative detectors are constantly being designed; New theories, methods, heating strategies and probe designs for estimating soil thermal properties, unfrozen water content and ice content are still in their infancy, but they may greatly improve our understanding of measurement techniques and the properties of partially frozen porous materials. For HP sensors and many other methods, such as TDR and neutron hygrometers, poor contact is a problem. Microscopically, the thermal contact resistance between the soil and the probe is affected by the contact area between them, and the contact area is affected by the size and geometry of the probe and the density of the soil particles, while the density of the soil particles is affected by the bulk density, the size distribution, the structure and the shape of the soil particles. In addition, available thermal conductivity models developed from steady-state measurements need to be evaluated and calibrated.

……”

And 

“……

In the rectangular coordinate system, this paper changes the previous cube model into spherical model, because in the soil’s spatial distribution, the spherical model is more in line with the general movement law of water and vapor in actual situation, while the edges and corners of the cube model have errors for the random flow of water. Therefore, this paper uses the spherical model to deduce water heat coupling equation. The choice of coordinate system is rectangular coordinate system, because the radius and angle of spherical coordinate system can be converted into the spatial correspondence of rectangular coordinate system, so this paper focuses on the minimum spatial physical unit of soil water movement, which is a spherical structure model.

……”

2. In Mathematical model, the mathematical calculation model of single-hole frozen soil column was proposed. The heat conduction equation, boundary conditions and Initial conditions were presented. However, the biggest problem is the phase transition of frozen soil. How do you consider the phase transition? This is the most important characteristic of frozen soil.

Answer: Yes, the phase change process of frozen soil directly affects the construction of water and heat transport model. By testing the physical parameters of frozen and unfrozen soil in severe cold area, and using the expression of soil and atmospheric natural temperature field, we analyze the influence of soil frozen and unfrozen physical parameters on soil natural temperature field at different depths, However, the phase transition process of frozen and unfrozen soil is not considered. Then, we use the coupled soil water and heat transfer equation, considering the changes of soil and atmospheric temperature field, establish the physical and mathematical model of soil heat transfer with phase change effect, and compare the numerical simulation results of whether phase change is considered in soil freezing process. At the same time, the impedance effect of ice is not considered in this paper.

The above is the analysis and thinking of frozen soil phase change in this paper, and the corresponding content has been described in the part of "one dimension Seasonal Frozen Soil Hydrothermal Coupling migration".

As following:

“……

, constructed water transport equation in frozen soil. Phase transition refers to the process of a substance changing from one phase to another. The physical and chemical properties of the substance system are completely the same. The homogeneous part with obvious interface with other parts is called phase, which corresponds to solid, liquid and gas states. The substance has solid phases, liquid phases and gas phases. The soil phase transition mainly refers to the heat absorbed or released by a certain amount soil, such as the soil micro unit in Fig.1, when it changes from one phase to another under certain conditions (such as constant temperature). There are three kinds of heat: evaporation heat (from liquid phase to gas phase), melting heat (from solid phase to liquid phase) and sublimation heat (from solid phase to gas phase). Lei Zhidong et al. [7] considered that the Richards equation only considered the effect of water migration and phase transition, but not the effect of heat conduction. However, the other scholars[9-16] considered the heat conduction and phase change latent heat, but did not consider the water migration. On the basis of these two equations, Lei Zhidong et al. [7] put forward the coupled water and heat transfer equation of frozen soil, which considered the factors of soil water transfer, heat conduction, water phase change latent heat and so on.

In this paper, the physical parameters of frozen and unfrozen soil in severe cold area are tested. The expression of natural temperature field of soil and atmosphere is used to analyze the influence of soil freezing and unfrozen physical parameters on soil’s natural temperature field at different depths, but the phase transformation process of soil freezing and unfrozen state is not considered.

…”

And

“……

The physical and mathematical models of soil heat transfer with phase change are established by using coupled soil water heat transfer equation, considering the changes of soil and atmospheric temperature field. The numerical simulation results of whether phase change is considered, and these results are compared and analyzed in soil freezing process. The results show that the curve of soil temperature with time is no longer smooth when there is phase change, and there is a gentle duration of soil temperature at the beginning of freezing and thawing, with which compared the soil without phase change; After freezing, the soil temperature with phase change is higher than that without phase change. With the increasing of depth, the temperature difference is increased. This conclusion will be proved in the later experiments. At the same time, the impedance effect of ice is not considered.

……”

3. This paper proposed a new hydrothermal coupling differential unit cell model that is based on a sphere. Compared with the traditional parallelepiped basic differential unit model, the proposed model has increased computational complexity. What are the advantages of this study?

Answer: Based on the cube model introduced in the previous literature, this paper proposes the physical process of water movement under the sphere model. The biggest advantage of this model is that the particle can do any trajectory movement in the 360 degree free space. As a water molecule, it can enter and exit the spherical structure designed in this paper in 360 degree free space.

In the past, the micro unit of cube can also analyze and study the movement of water molecules, but there are "edges" and "angles" in the cube, so it is not easy to calculate the movement of water molecules when they pass through the edges and angles, and the sphere structure has no "edges" and "angles", which fully ensures that water can move in any direction, which is more in line with the actual situation, Therefore, this paper proposes such a new spherical micro unit for induction, which is the basis of analyzing the law of water movement.

The above content is added to the corresponding position of first natural paragraph in the part of mathematical model � A. seasonal frozen soil water movement.

4. Based on local meteorological data, the physical experimental was carried out. But the detailed engineering geological conditions, hydrogeological conditions and construction freezing process are not introduced. This is the key. The reliability and discreteness of the result are doubtful.

Answer: Thank you for your comment. It's very pertinent. We focus on revising this opinion during this period. The experimental contents of data acquisition, such as the experimental process of data acquisition, the introduction of sensors and the arrangement of acquisition devices on site are added. The previous tests are reorganized. The engineering geological conditions, hydrogeological conditions and construction freezing process of local environment are described. At the same time, the collected data are uploaded in the form of attachments in the system, the paper’s following content is to analyze and fit the historical data collected, finally make the change analysis of water and heat migration in frozen soil.

Please review for the above modifications, which are in the "material and methods" section.

---

## [Decision Letter · Decision Letter 1]

27 Aug 2021

PONE-D-21-16032R1

A new seasonal frozen soil water-thermal coupled migration model and its numerical simulation

PLOS ONE

Dear Dr. Sun,

Thank you for submitting your manuscript to PLOS ONE. After careful consideration, we feel that it has merit but does not fully meet PLOS ONE’s publication criteria as it currently stands. Therefore, we invite you to submit a revised version of the manuscript that addresses the points raised during the review process.

We look forward to receiving your revised manuscript.

Kind regards,

Jianguo Wang, PhD

Academic Editor

PLOS ONE

Journal Requirements:

Additional Editor Comments (if provided):

I have one comment on English and presentations. They should be improved significantly to meet the requirements for publication. For example, I read the Abstract of manuscript and feel that some expressions are not smooth or conventional or unclear.

Reviewers' comments:

Reviewer's Responses to Questions

**Comments to the Author**

1. If the authors have adequately addressed your comments raised in a previous round of review and you feel that this manuscript is now acceptable for publication, you may indicate that here to bypass the “Comments to the Author” section, enter your conflict of interest statement in the “Confidential to Editor” section, and submit your "Accept" recommendation.

Reviewer #1: All comments have been addressed

Reviewer #2: All comments have been addressed

2. Is the manuscript technically sound, and do the data support the conclusions?

Reviewer #1: Yes

Reviewer #2: Yes

3. Has the statistical analysis been performed appropriately and rigorously? 

Reviewer #1: Yes

Reviewer #2: Yes

4. Have the authors made all data underlying the findings in their manuscript fully available?

Reviewer #1: Yes

Reviewer #2: Yes

5. Is the manuscript presented in an intelligible fashion and written in standard English?

Reviewer #1: Yes

Reviewer #2: Yes

6. Review Comments to the Author

Reviewer #1: I have no further questions, it has improved greatly right now. The manuscript can be accepted as it is. In future, I am expected to see the use of the model to predict the water-heat behaviors of soils at field.

Reviewer #2: The revisions are satisfactory to this reviewer. The manuscript is recommended for publication in its current form.

7. PLOS authors have the option to publish the peer review history of their article (what does this mean?). If published, this will include your full peer review and any attached files.

Reviewer #1: No

Reviewer #2: No

---

## [Author Response · Author response to Decision Letter 1]

16 Sep 2021

We thank editors for reviewing this article, they put forward some very constructive opinions. We answer these questions one by one, the details are as follows:

Additional Editor Comments (if provided):

I have one comment on English and presentations. They should be improved significantly to meet the requirements for publication. For example, I read the Abstract of manuscript and feel that some expressions are not smooth or conventional or unclear.

Answer: Thank you very much for your advice.

After the last round of revision, the article has been greatly adjusted, mainly includes the sections of abstract, introduction and conclusion. The contents and words of these three parts are almost re-written, so we have translated the contents of these three parts again, after many modifications, the current version is formed.

The other sections of the article is that we hired American AJE company（https://www.aje.cn/)）, the translation and polishing have been translated by professional institutions, so there is little change.

Thank the editor again for carefully reviewing!

We thank the reviewers for reviewing this article, and they put forward some very constructive opinions. We answer the reviewers' questions one by one, the details are as follows:

Reviewer #1: I have no further questions, it has improved greatly right now. The manuscript can be accepted as it is. In future, I am expected to see the use of the model to predict the water-heat behaviors of soils at field.

Answer: Thank you very much for your pertinent and valuable comments. We have carefully revised the manuscript according to your comments to form the current form. We also hope to see the research reports on soil physics published in the future.

Reviewer #2: The revisions are satisfactory to this reviewer. The manuscript is recommended for publication in its current form.

Answer: Thank you for your affirmation of our work, and your careful review of this article. Your suggestions are very helpful to the improvement of our work. We also hope to see the latest achievements in soil physics in the future.

---

## [Editor Report · Decision Letter 2]

7 Oct 2021

A new seasonal frozen soil water-thermal coupled migration model and its numerical simulation

PONE-D-21-16032R2

Dear Dr. Sun,

We’re pleased to inform you that your manuscript has been judged scientifically suitable for publication and will be formally accepted for publication once it meets all outstanding technical requirements.

Kind regards,

Jianguo Wang, PhD

Academic Editor

PLOS ONE
---

## [Editor Report · Acceptance letter]

11 Oct 2021

PONE-D-21-16032R2 

A new seasonal frozen soil water-thermal coupled migration model and its numerical simulation 

Dear Dr. Sun:

I'm pleased to inform you that your manuscript has been deemed suitable for publication in PLOS ONE. Congratulations! Your manuscript is now with our production department. 

Kind regards, 

on behalf of

Dr. Jianguo Wang 

Academic Editor

PLOS ONE